# RANKING-AWARE ADAPTER FOR TEXT-DRIVEN IMAGE ORDERING WITH CLIP

**Wei-Hsiang Yu**[1]    **Yen-Yu Lin**[1]    **Ming-Hsuan Yang**[2]    **Yi-Hsuan Tsai**[3]
[1]National Yang Ming Chiao Tung University    [2]UC Merced    [3]Atmanity Inc.

## ABSTRACT

Recent advances in vision-language models (VLMs) have made significant progress in downstream tasks that require quantitative concepts such as facial age estimation and image quality assessment, enabling VLMs to explore applications like image ranking and retrieval. However, existing studies typically focus on the reasoning based on a single image and heavily depend on text prompting, limiting their ability to learn comprehensive understanding from multiple images. To address this, we propose an effective yet efficient approach that reframes the CLIP model into a learning-to-rank task and introduces a lightweight adapter to augment CLIP for text-guided image ranking. Specifically, our approach incorporates learnable prompts to adapt to new instructions for ranking purposes and an auxiliary branch with ranking-aware attention, leveraging text-conditioned visual differences for additional supervision in image ranking. Our ranking-aware adapter consistently outperforms fine-tuned CLIPs on various tasks and achieves competitive results compared to state-of-the-art models designed for specific tasks like facial age estimation and image quality assessment. Overall, our approach primarily focuses on ranking images with a single instruction, which provides a natural and generalized way of learning from visual differences across images, bypassing the need for extensive text prompts tailored to individual tasks. The source code is available at https://github.com/uynaes/RankingAwareCLIP

## 1 INTRODUCTION

Human perception can handle multiple images simultaneously, performing tasks like semantic categorization, e.g., counting cats in an image, and abstract evaluations, e.g., image quality, aesthetics, and facial age. This ability is critical for comprehending the relationships between objects and concepts across images and is essential to effective reasoning and decision-making. In machine learning, learning-to-rank (LTR) algorithms are designed to investigate these relationships among items based on their relevance to a given context. LTR has shown remarkable success in various domains, including information retrieval (Chen et al., 2021) and document ranking (Cao et al., 2007).

The emergence of Vision-Language Models (VLMs) like CLIP (Radford et al., 2021) has sparked a growing interest in adapting pre-trained VLMs to understand specific image attributes (Ke et al., 2023; Liang et al., 2023; Wang et al., 2023a;d). However, prior research has primarily focused on single-image reasoning (see Figure 1a), potentially hindering the model's ability to explore complex relationships across images. In this paper, we aim to equip the pre-trained CLIP to rank multiple images based on various text queries, thereby facilitating a wide range of tasks from semantic understanding (e.g., object counting) to abstract attribute estimation (e.g., image quality assessment). To this end, we introduce a ranking-aware adapter that aims to provide text-visual embeddings with a sense of visual relevance to a given text query, which is lightweight and effective for learning image ordering tasks (see Figure 1b).

A straightforward approach to adapt VLMs to a target task is to conduct instruction tuning with pre-defined text queries followed by post-processing for ranking. For instance, Paiss et al. (2023) pre-define the text query to include the number of objects like "three cats" for each image. In this manner, the CLIP model can be fine-tuned using contrastive objectives for counting, and then images are sorted based on their predicted counts. Similarly, Zhang et al. (2022) effectively addresses the depth estimation task by defining an object's proximity from "close" to "far." Other approaches augment the CLIP text encoder with additional learnable prompts to capture ordinal relationships

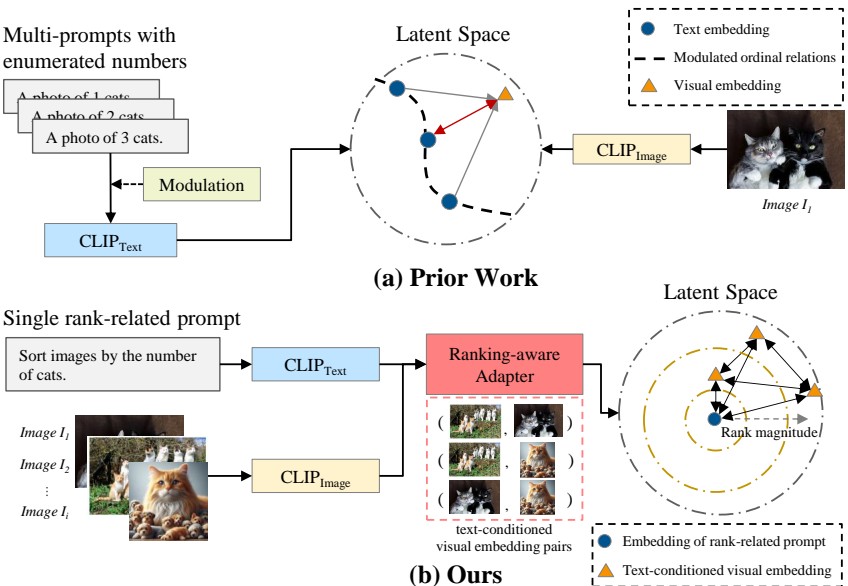

Figure 1: **Comparisons with prior work**. (a) Prior works require generating caption combinations covering numbers of bins from $N_1, N_2, ..., N_i$ with the task-related target, such as "cat," paired with each image (e.g., $I_1, I_2$). In addition, text modulation is necessary to map these numerical values into an ordinal latent space for contrastive learning. (b) Our method streamlines the ranking process through a learning-to-rank framework. A pre-trained CLIP model encodes images and a single rank-related prompt. A lightweight, ranking-aware adapter then generates text-conditioned visual embedding pairs and their relational differences. By optimizing the relational differences among pairs, our approach learns the visual relevance to the given text query.

and align these prompts with image embeddings (Li et al., 2022; Wang et al., 2023d). However, aligning visual embeddings with text embeddings for ranking may be suboptimal since it is difficult to enumerate and learn all prompts to represent the spectrum of ranks, especially for data in long-tailed distributions. While some methods achieve satisfactory results through task-specific pre-training and custom decoders, their specialized decoders often limit their adaptability to other ranking tasks.

To address the aforementioned challenges, we propose an LTR framework incorporating a lightweight ranking adapter for the pre-trained CLIP model. This framework involves: 1) Transforming the paired image-text contrastive objective to an LTR objective by directly pairing a list of images with their scores alongside text guidance; 2) Introducing a ranking-aware adapter comprising a regression branch for ranking score prediction and an ordinal branch for pairwise data order supervision; and 3) Developing an attention mechanism to extract relative feature responses for a pair of images, thereby constructing relative visual embeddings conditioned by queried text prompts.

We evaluate our method on four tasks spanning diverse numerical concepts, including facial aging estimation (Samek et al., 2017), object count sorting (Singh et al., 2024), image quality/aesthetics assessment (Hosu et al., 2020; Murray, Naila and Marchesotti, Luca and Perronnin, Florent, 2012), and dating historical colored images (Palermo et al., 2012). Our approach consistently performs favorably against CLIP baselines and state-of-the-art methods in terms of ranking and retrieval qualities, even though these competing methods are fine-tuned for target tasks. Instead of focusing on enhancing CLIP's text encoder, our method unifies different ranking tasks in one framework followed by a lightweight adapter with a ranking-aware attention module to capture text-conditioned image differences, directly contributing to learning the distinction across ranking scores.

The main contributions of this work are as follows: 1) We introduce a general ranking-related text prompt and integrate learning-to-rank into the CLIP model using a lightweight adapter for multi-image ranking. 2) We propose a ranking-aware attention module to facilitate feature learning for the ranking purpose across images conditioned on the given text. 3) Without requiring additional task-specific pre-training, we validate our method on various ranking tasks in benchmark datasets against fine-tuned CLIP baselines and other task-specific approaches, providing insight toward a unified framework for image ranking tasks.

## 2 RELATED WORK

**Contrastive vision-language models.** Learning and aligning multiple modality-specific representations from large-scale image-text datasets have significantly advanced the performance of numerous multi-modal tasks (Alayrac et al., 2022; Chen et al., 2023; Kim et al., 2021; Pham et al., 2023; Radford et al., 2021; Singh et al., 2022; Yu et al., 2022; Lin et al., 2024). CLIP (Radford et al., 2021), a pioneering vision-language model (VLM), aligns visual and textual encoders via a pairwise contrastive objective and excels at zero-shot classification. Due to their robust and general representations, VLMs are widely adopted as foundational models for a variety of downstream tasks, such as segmentation (Lüddecke & Ecker, 2022; Rao et al., 2022), detection (Wu et al., 2023), and dense prediction (Auty & Mikolajczyk, 2023; Hu et al., 2024). While CLIP effectively captures global information, its training objective, matching image content to sentences, hinders its ability to discern relationships across images. Several approaches are proposed to address this issue, such as task-dependent decoder training (Jiang et al., 2023; Ke et al., 2023) and fine-tuning with task-specific image-caption pairs (Liang et al., 2023; Paiss et al., 2023; Wang et al., 2023a;b). While these approaches augment VLMs with additional relational knowledge, they often necessitate task-specific modifications to model architectures or data preprocessing, limiting their generalizability to diverse downstream tasks.

In this paper, we introduce a framework that empowers CLIP to rank a set of images based on the attribute specified by a given text, such as ranking images based on object counts, facial aging, or visual qualities, without requiring any task-specific pre-processing or model modifications. Our method enables the model to learn differences from randomly paired images and correlate them with their ranking distances, thereby enhancing ranking performance and achieving promising results compared to task-specific methods.

**Learning-to-rank.** Learning-to-rank (LTR) is pivotal in training models to effectively sort a list of items, making it essential to information retrieval and recommendation systems that strive to present the most relevant items in response to queries (Cao et al., 2007). LTR has a broad spectrum of applications, such as person/face re-identification (Chen et al., 2017; 2021), artwork retrieval (Yemelianenko et al., 2023), and storytelling evaluation (Hsu et al., 2022). LTR models are typically trained using ranked lists of items, associated queries, and relevance scores, such as click-through data, user preference scores, or similarities. Ranking objectives such as pairwise Hinge loss, triplet loss (Schroff et al., 2015), or listwise ranking loss (Burges et al., 2005) are commonly employed to optimize LTR models. While classification predicts discrete categories and regression predicts continuous values for individual items, LTR focuses on cross-item comparisons based on their relative relevance to a query. This emphasis on relative relevance encourages models to learn quantitative representations that distinguish between items based on a given query.

Recent works like OrdinalCLIP (Li et al., 2022), L2RCLIP (Wang et al., 2023d), and the concurrent work NumCLIP (Du et al., 2024) combine CLIP with LTR for ordinal regression tasks such as facial age estimation and image dating. They focus on designing the text encoder to map numerical labels to a continuous space for improved image-text alignment. In this work, we also combine CLIP and LTR for ranking tasks but focus on optimizing text-conditioned image embedding distances between paired images to enhance ranking, a promising yet unexplored area in visual understanding.

## 3 PROPOSED METHOD

This paper proposes a method to equip a pre-trained VLM, e.g., CLIP, with ranking a set of images based on a given textual query, such as one particular object count, abstract concept, or subjective preference. Figure 2 provides an overview of our framework, where we formulate this task as a learn-to-rank problem and derive a lightweight adapter with a ranking-aware attention module to extract relative feature responses across images so that the model can rank images.

### 3.1 LEARNING-TO-RANK FORMULATION

A straightforward approach to teach VLMs for image scoring based on textual queries is to yield prompts by associating images with potential numerical indices, such as "A photo of [0, 1, 2, 3, ...] cats." for object counting (Paiss et al., 2023) or "A photo of [1, 2, 3, ...] years old face." for age estimation (Wang et al., 2023d). However, enumerating all numerical indices is impractical and computationally infeasible when dealing with large or

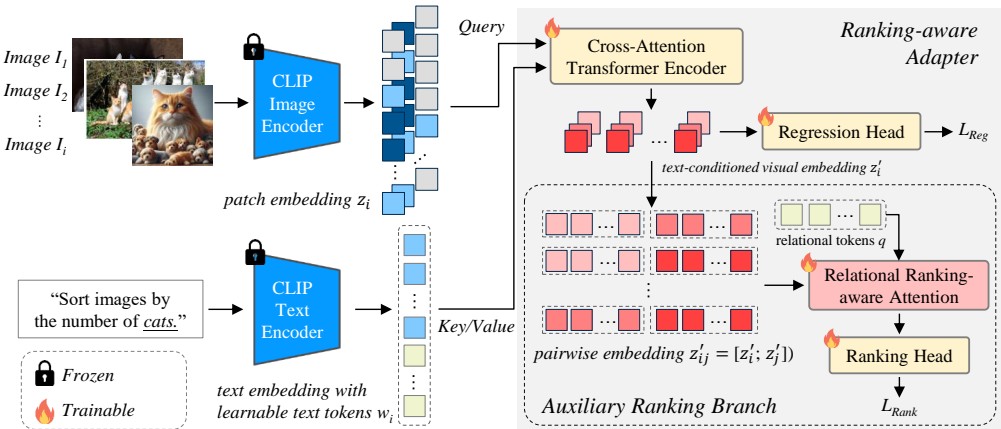

Figure 2: **Framework overview**. We encode the given images and the query caption using the pre-trained CLIP model. The proposed ranking adapter compiles text-conditioned visual embedding $\{z_i'\}$ via the transformer cross-attention mechanism, where patch embeddings $\{z_i\}$ serve as queries and text embeddings $\{w_i\}$ act as key-value pairs. The ranking adapter comprises two heads: the regression and ranking heads. The former predicts image ranking scores using features of individual images. The latter employs a ranking-aware attention mechanism to explore relative feature responses across images upon which these images are ranked.

even uncertain index ranges. Furthermore, tasks like object counting and facial age estimation frequently exhibit long-tail data distributions, making discrete captions less balanced. For continuous scoring targets like image preference or colorfulness estimation, discretizing scores into bins can be helpful (Hu et al., 2024; Auty & Mikolajczyk, 2023). However, determining an appropriate number of bins and cutoffs remains challenging and may vary depending on the applications.

To address these challenges of ranking images based on text guidance, we design a universal instruction template for all images containing keywords such as "Sort images by the number of [cats]." and use their query-related values as our learning targets. Then, we adopt an objective to optimize the model for image ranking rather than pairing each image with all potential text queries.

## 3.2 RANKING ADAPTER WITH RELATIONAL ATTENTION

The proposed method aims to augment CLIP for sorting images based on a given textual expression, where our model comprises a frozen CLIP backbone and a lightweight adapter. The adapter is composed of a cross-attention encoder and two decoders: one is a regression head for predicting single-image ranking scores, and the other is derived with pairwise supervision for cross-image ranking. Figure 2 depicts our proposed method.

**Vision-language backbone.** We employ a frozen pre-trained CLIP model as our feature extractor. To accommodate various tasks that may require either local or global image features (such as sorting by object counts or image brightness), we discard the pooling layers as suggested in Paiss et al. (2023) and Rao et al. (2022). This retains the image token embedding $z_i \in \mathbb{R}^{p \times d}, i \in \{1, 2, ..., B\}$, where $P = 100$ denotes the number of patch tokens and $B$ is the batch size, and the text embedding $w_i \in \mathbb{R}^{t \times d}$, where $t = 77$ is the number of text tokens. Both image and text embeddings share a latent dimension of $d = 768$. Besides, given that our text prompts are formulated for ranking, which differs from the original captions, we append the additional $t' = 32$ text tokens to adapt to the ranking-related text prompts.

**Adapting CLIP to image sorting with the ranking-aware adapter.** Inspired by Yu et al. (2022), we utilize cross-attention to integrate textual information into image tokens to enhance their contextual richness for accurate image ranking. Specifically, each encoding block in our design includes a self-attention layer for image tokens, followed by a cross-attention layer where image tokens serve as queries and text tokens as key-value pairs. Linear layers are then used for embedding projection $\mathbb{R}^d \mapsto \mathbb{R}^{d'}$, where $d' = 512$. This structure is repeated twice to effectively fuse text and image embeddings, resulting in the text-conditioned visual embedding $z_i'$. Subsequently, the output

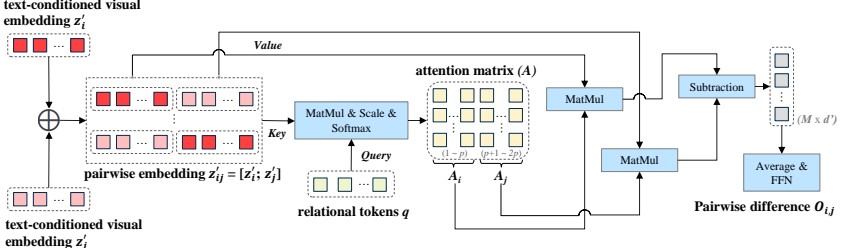

Figure 3: **Relational ranking-aware attention.** First, the text-conditioned visual embeddings of two images $\{z_i'\}$ and $\{z_j'\}$ are concatenated to form a pairwise embedding $\{z_{ij}'\}$. This pairwise embedding is used as the key in the attention mechanism, with relational tokens $\{q\}$ serving as the query. The resulting attention matrix $A$ is split into two parts, $\{A_i\}$ and $\{A_j\}$, corresponding to the attention assigned to each image. Using these matrices, the attention outputs, $\{O_i\}$ and $\{O_j\}$ are computed with $\{z_i'\}$ and $\{z_j'\}$ as values. Finally, the pair's difference is computed through subtraction, averaged over relational tokens, and processed by an FFN to generate the pairwise difference $O_{i,j}$.

embedding $z_i'$ is routed to two parallel branches. The first branch is a feed-forward sub-network (FFN) regressor that directly predicts the ranking score. The second branch is an auxiliary ranking module with a relational-aware attention mechanism, which characterizes the discrepancies between an image pair, aligning with their corresponding ranking score differences.

**Relational ranking-aware attention.**     Given a pair of images, this attention mechanism is designed to identify and highlight the visual discrepancies that correlate with their respective ranking scores. As depicted in Figure 3, the visual-text embeddings of two images are concatenated to form a pairwise embedding, which is subsequently processed through an attentional pooling (Lee et al., 2019).

Specifically, given two images $I_i$ and $I_j$, we concatenate their embeddings, $z_i'$ and $z_j'$, from the cross-attention transformer encoder to yield a pairwise representation: $z_{ij}' = [z_i'; z_j'] \in \mathbb{R}^{2p \times d'}, ij \in \{1, 2, ..., B^2\}$, where $p$ is the number of patch tokens, and $d'$ is the embedding dimension.

We then initialize the relational tokens $q \in \mathbb{R}^{M \times d'}$, where $M$ is the number of tokens, as the queries. The concatenated embedding $z_{ij}'$ acts as key-value pairs. This allows us to generate relational representations. The attention matrix $A$ is calculated using scaled dot-product attention via

$$A = \texttt{Softmax}(\frac{q \cdot (k_i \oplus k_j)^T}{\sqrt{d'}}) = \texttt{Softmax}(\frac{q \cdot K^T}{\sqrt{d'}}), \tag{1}$$

where $K = z_{ij}' = [z_i'; z_j'] \in \mathbb{R}^{2p \times d'}$ acts as keys. After computing $A$, it is divided into two parts: $A_i$ and $A_j$, corresponding to the attention applied to $z_i'$ and $z_j'$, respectively. Here, $z_i'$ and $z_j'$ serve as values, denoted as $V_i$ and $V_j$ for the second dot-product operation:

$$O_i = A_i \cdot V_i \quad \text{and} \quad O_j = A_j \cdot V_j. \tag{2}$$

Finally, the relative difference between the outputs, $O_i$ and $O_j$, is computed:

$$O_{i,j} = FFN(\frac{\sum_{m=1}^{M}(O_{i,m} - O_{j,m})}{M}), \tag{3}$$

where $O_{i,m}$ is aggregated responses from image $i$ with the $m$-th relational token. The pairwise difference $O_{i,j}$ is averaged over $M$ relational tokens and passed through a feed-forward network, yielding an one-dimensional value for the pairwise ranking objective.

Unlike conventional attention mechanisms that process images individually, our proposed relational attention aggregates distinct features between a pair of images. The resulting features capture differences between images and hence are discriminative, leading to more accurate ranking predictions.

**Training objective and loss functions.**     To optimize the model for both accurate ranking and score prediction, we combine two complementary loss functions: a regression loss and a pairwise ranking loss. The regression branch predicts the ranking score for each image individually, and we employ Smooth L1 loss to minimize the difference between the predicted score $s_i$ and the ground truth $y_i$:

$$L_{reg} = \begin{cases} \frac{1}{2}(y_i - s_i)^2, & \text{if } |y_i - s_i| < 1, \\ |y_i - s_i| - 0.5, & \text{otherwise.} \end{cases} \tag{4}$$

This encourages precise alignment between predicted scores and the query-based ground truth values, particularly penalizing large errors.

The pairwise ranking branch focuses on relative ordering, using a pairwise ranking loss to ensure that for any two images, e.g., when $y_i > y_j$, the model is is derived by the following loss function:

$$L_{rank}(y, O_{i,j}) = \sum_{\{(i,j)|y_i > y_j\}} \max(0, 1 - O_{i,j}). \tag{5}$$

This loss directly optimizes the model's ability to capture the correct relative differences between image pairs, which is crucial for ranking tasks.

Finally, we combine the two objectives into a unified loss function:

$$L = \alpha L_{reg} + L_{rank}. \tag{6}$$

where $\alpha$ is a hyperparameter (set to $0.2$ in our experiments) to balance the importance of the regression and ranking objectives. By jointly optimizing both individual scores and pairwise ranking consistency, our method ensures robust performance across various ranking tasks.

## 4 EXPERIMENTAL RESULTS

### 4.1 TASKS AND DATASETS

**Facial age estimation.** Facial age estimation predicts the age of a given face. We use the Adience dataset (Samek et al., 2017), which includes $13,027$ images labeled across 8 age groups, following the data split from Wang et al. (2023d). We use the text prompt "`sort images by the person's age group.`" for this task.

**Historical colored image dating.** The historical colored image dataset (Palermo et al., 2012) is a widely-used benchmark for predicting the decade of a given historical colored image, consisting of $1,325$ images labeled across 5 decades ranging from the 1930s to the 1970s. We follow the standard ordinal regression setting as that in Wang et al. (2023d). The text prompt we use is "`sort images by the taken date of the photo.`" for this task.

**Image quality and aesthetics assessment.** For ranking images based on subjective preference and objective properties, we employ the KonIQ-10k dataset (Hosu et al., 2020) for assessing image qualities and the Aesthetic Visual Analysis (AVA) dataset (Murray, Naila and Marchesotti, Luca and Perronnin, Florent, 2012) for evaluating image aesthetics, using the task prompt "`sort images by the image quality of [Property].`". In addition to user voting or mean opinion score (MOS), we include other image quality attributes such as colorfulness, contrast, and brightness. We evaluate models using the official splits, which have $2,015$ and $19,930$ test images in the KonIQ-10k and AVA datasets, respectively.

**Object count sorting.** We evaluate the performance of scoring images according to the quantity of queried objects using the text prompt instruction, "`sort images by the number of [Category].`". The COCO-REM dataset (Singh et al., 2024), an annotation revised version of the COCO dataset, serves as the test bed. We work on the 80 categories of the COCO dataset, comprising $118,287$ training and $5,000$ testing images.

**Evaluation metrics.** We evaluate facial age estimation and historical image dating using mean absolute error (MAE) and accuracy for comparison with prior works. We assess performance using Pearson's linear correlation coefficient (PLCC) and Spearman's rank correlation coefficient (SRCC) for image quality assessment and object count sorting.

### 4.2 EXPERIMENTAL SETTING

We use CLIP with ConvNeXt-L, containing a ConvNext-Large (Liu et al., 2022) image encoder with the image resolution set to $320 \times 320$ and the hidden dimension to $d_1 = 768$, as well as a transformer-based text encoder of 12 layers with $w = 77$ text tokens and the hidden dimension set to $d_2 = 768$. The model weights pre-trained on the LAION-5B dataset (Schuhmann et al., 2022) are adopted and frozen for both the image and text encoders. To adapt our model to the given text, we concatenate learnable prompts $s'$ of size 32 to the text embedding produced by the CLIP text

Table 1: **Results on facial age estimation and historical colored image dating.** The mean and standard deviation are listed. We take numbers from Wang et al. (2023d) for the results of the reference methods.

| Method | Adience | | HCI | |
|---|---|---|---|---|
| | Accuracy (%) | MAE | Accuracy (%) | MAE |
| Zero-shot CLIP | 43.3 (3.6) | 0.80 (0.02) | 26.1 (0.6) | 1.48 (0.03) |
| CoOp | 60.6 (5.5) | 0.50 (0.08) | 51.9 (2.6) | 0.76 (0.06) |
| OrdinalCLIP | 61.2 (4.2) | 0.47 (0.06) | 56.4 (1.7) | 0.67 (0.03) |
| L2RCLIP | **66.2** (4.4) | **0.36** (**0.05**) | 67.2 (1.6) | 0.43 (0.03) |
| NumCLIP | - | - | 69.6 (2.0) | 0.35 (0.03) |
| InstructBLIP | 63.7 | 0.41 | 30.9 | 0.96 |
| Ours | 65.2 (2.9) | **0.36** (**0.03**) | **72.8** (**2.6**) | **0.32** (**0.03**) |

Table 2: **Results on object count sorting.** For BLIP-2, Flamingo, InstructBLIP, and VLM-VLIA, we follow the text prompts for object counting as specified in their studies.

| Method | Fine-tuning | PLCC (↑) | SRCC (↑) |
|---|---|---|---|
| BLIP-2 | | 0.284 | 0.252 |
| Flamingo (10-shot) | | 0.033 | 0.031 |
| InstructBLIP | | 0.509 | 0.485 |
| VLM-VILA | | 0.558 | 0.507 |
| Zero-shot CLIP | | 0.026 | 0.001 |
| CountingCLIP Paiss et al. (2023) | ✓ | 0.251 | 0.422 |
| Ours | ✓ | **0.624** | **0.557** |

Table 3: **Results on the IQA/IAA task**. We report the PLCC and SRCC of the mean opinion score (*MOS*) to compare with baselines and the state-of-the-art methods. Note that MUSIQ (Ke et al., 2021) only serves as the reference purpose since it is a purely vision-based method optimized for this task. We take numbers from Ke et al. (2023) for the results of the reference methods.

| Method | Task-related pertaining | Fine-tuning | KonIQ-10k | | AVA Dataset | |
|---|---|---|---|---|---|---|
| | | | PLCC (↑) | SRCC (↑) | PLCC (↑) | SRCC (↑) |
| **Purely vision-based (task-specific)** | | | | | | |
| MUSIQ Ke et al. (2021) | | ✓ | 0.924 | 0.937 | 0.726 | 0.738 |
| **VLM-based (task-specific)** | | | | | | |
| VILA-P Ke et al. (2023) | ✓ | | - | - | 0.657 | 0.663 |
| VILA-R Ke et al. (2023) | ✓ | ✓ | **0.919** | **0.932** | **0.774** | **0.774** |
| CLIP (fine-tuned) | | ✓ | 0.245 | 0.216 | 0.162 | 0.160 |
| InstructBLIP Dai et al. (2023) | | | 0.211 | 0.163 | 0.229 | 0.226 |
| CLIP-IQA Wang et al. (2023c) | | | 0.695 | 0.727 | 0.420 | 0.415 |
| CLIP-IQA+ Wang et al. (2023c) | | ✓ | 0.895 | 0.909 | 0.677 | 0.587 |
| Hentschel et al. (2022) | | ✓ | - | - | 0.731 | 0.741 |
| Ours | | ✓ | **0.919** | **0.911** | **0.760** | **0.747** |

encoder. Following Jiang et al. (2023), we discard the pooling and projection of the visual encoder, leaving $p = 100$ patch tokens as inputs to our ranking adapter. The ranking adapter consists of a cross-attention transformer encoder with two encoding blocks, followed by two parallel branches: 1) a regression head with two multi-layer perceptrons (MLP) blocks and 2) a ranking head with contrastive ranking-aware attention using $M = 16$ relational tokens and three MLP blocks.

We take a random horizontal flipping as data augmentation and resize the images to $320 \times 320$ without cropping. We implement the ranking adapter upon the OpenCLIP (Ilharco et al., 2021) framework and optimize the model using a combination of Smooth L1 loss and Hinge loss with an AdamW optimizer at a learning rate of $1e-5$, weight decay of $0.01$, and batch size of $64$. We use 220k steps for object counts sorting and 144k steps for image-quality assessment, facial age estimation, and dating historical image tasks. We conduct all experiments on one NVIDIA RTX-3090Ti GPU.

## 4.3 QUANTITATIVE RESULTS

In the following, we evaluate the efficacy of our method by comparing it with baseline models on four quantitative tasks with various challenges.

**Facial age estimation and historical colored image dating.** We evaluate our method against zero-shot CLIP and four existing approaches for facial age estimation and historical color image (HCI) dating. The benchmarks include CoOp (Zhou et al., 2022), OrdinalCLIP (Li et al., 2022), L2RCLIP (Wang et al., 2023d), and the concurrent NumCLIP (Du et al., 2024). As shown in Table 1, our approach, which leverages visual differences between images, achieves competitive performance on facial age estimation in the Adience dataset, rivaling methods focusing on text embedding modulation. Note that while our method performs slightly lower than L2RCLIP, the standard deviation indicates our approach's stability over L2RCLIP. We also validate our model on the UTKFace dataset, where it outperforms other approaches (see Table 7 in Appendix A.1). Moreover,

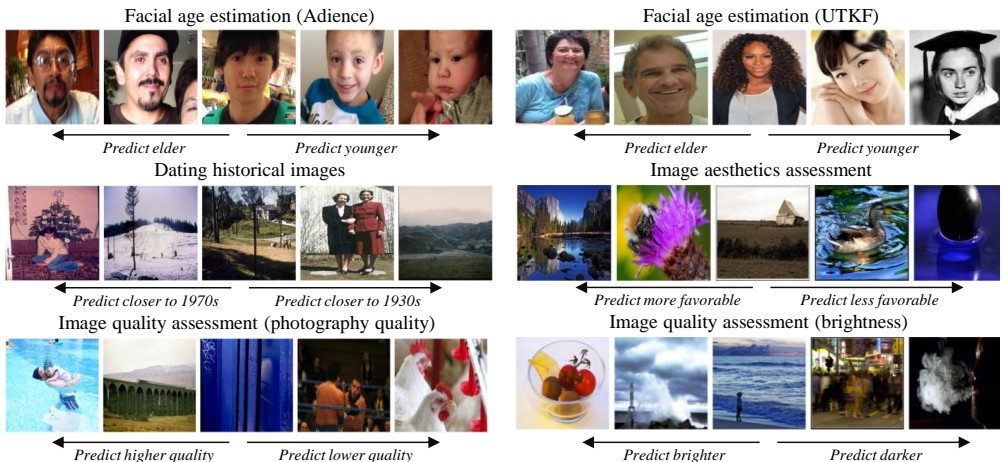

Figure 4: **Qualitative examples of our model.** We visualize the ranking performance on facial age estimation, dating historical images, and image quality and aesthetics assessment.

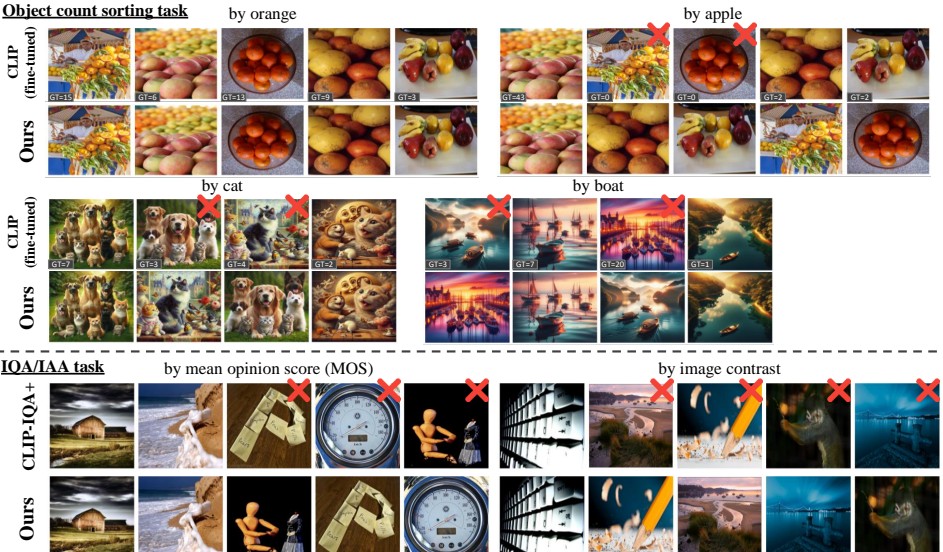

Figure 5: **Qualitative examples on object count sorting and IQA/IAA**. The images are sorted from highest to lowest score according to the textual cues. The red cross (×) represents the wrong sorting position in the list. AI artworks are generated by DALL·E 3 (Betker et al., 2023). Our method accurately ranks images from simple to complex compositions with multiple object categories in real photos and artworks. For image sorting based on quality properties, our method outperforms fine-tuned CLIP-IQA (CLIP-IQA+).

our method records an MAE of 0.32 on the HCI dataset, surpassing the existing state-of-the-art by a margin of 0.04 (more than 9% improvement). Figure 4 further demonstrates our method's ability to accurately rank images in various scenarios.

**Object count sorting.** For object count sorting, we compare our method with two CLIP baselines: the zero-shot CLIP model and fine-tuned Count-to-Ten (Paiss et al., 2023), along with four representative VLM models — BLIP-2 (Li et al., 2023b), Flamingo (Alayrac et al., 2022), InstructBLIP (Dai et al., 2023), and VLM-VILA (Lin et al., 2024) — using visual-question-answering (VQA) protocol. While the two CLIP baselines rely on image-text pairs to determine counts, with object numbers in COCO-REM ranging from one to hundreds, we combine categories and numbers from 0 to 100 in the caption during fine-tuning and inference, e.g., "A photo of {7} {cat}." For the VLM benchmarks, we follow the text prompts for object counting as specified in their studies and parse the number from the generated caption. As reported in Table 2, our proposed method achieves a PLCC

Table 4: **Results of other image attributes on the AVA dataset.**

|  | MOS | | Contrast | | Colorfulness | | Brightness | |
|---|---|---|---|---|---|---|---|---|
|  | PLCC ($\uparrow$) | SRCC ($\uparrow$) | PLCC ($\uparrow$) | SRCC ($\uparrow$) | PLCC ($\uparrow$) | SRCC ($\uparrow$) | PLCC ($\uparrow$) | SRCC ($\uparrow$) |
| CLIP (fine-tuned) | 0.162 | 0.160 | $-0.058$ | $-0.071$ | 0.289 | 0.308 | 0.087 | 0.061 |
| CLIP-IQA | 0.420 | 0.415 | 0.227 | 0.210 | 0.472 | 0.486 | 0.433 | 0.442 |
| CLIP-IQA+ | 0.677 | 0.687 | 0.695 | 0.712 | 0.881 | 0.895 | 0.908 | 0.919 |
| Ours | **0.760** | **0.747** | **0.951** | **0.946** | **0.973** | **0.966** | **0.981** | **0.979** |

of $0.624$ and an SRCC of $0.557$, which is significantly better than the original CLIP, its task-specific fine-tuned variant, and large VLMs using the VQA protocol. Notably, our method improves by eliminating the need to exhaustively enumerate object-count combinations during training and inference, typically required in pairwise contrastive approaches.

In addition, our method supports training with multiple queries simultaneously, such as sorting different objects (e.g., cat, dog, boat). Most CLIP-based methods struggle with this because they require pairing numbers and queried objects simultaneously, leading to overly complex text prompts. Figure 5 shows that our method can rank images based on the queried category for real and synthesized images containing objects of one or multiple classes.

**Image quality and aesthetics assessment.** We compare our method to three existing methods for image quality and aesthetic assessment (IQA/IAA), including the CLIP model, CLIP-IQA (Wang et al., 2023c), and VILA (Ke et al., 2023), on two benchmarks: the KonIQ-10k and AVA datasets. In Table 3, our method performs favorably against CLIP-IQA and its fine-tuned variant, CLIP-IQA+, while maintaining competitive performance compared to the VILA variants that require pretraining on task-related datasets on the AVA caption dataset (Ghosal et al., 2019) (i.e., VILA-P) and fine-tuned with a rank adapter (i.e., VILA-R) (see Figure 5 for more visual comparisons with CLIP-IQA+).

Moreover, we explore our model's capacity to score images based on various image attributes, including contrast, colorfulness, and brightness. We compare the performance with CLIP and CLIP-IQA using the AVA dataset. To fine-tune CLIP, we discretize attribute values into bins and pair images with multiple captions describing the specific attribute values. For example, the caption "`The contrast of the image is higher than 0.1 and lower than 0.2.`" matches an image with a contrast of $0.15$. For CLIP-IQA, we employ antonym prompts, such as "Good photo." versus "Bad photo." for MOS, "Colorful photo." versus "Dull photo." for colorfulness, etc., to train the model on each image attribute using the official repository from Wang et al. (2023c).

In Table 4, our proposed method consistently outperforms task-specific fine-tuned CLIP and CLIP-IQA+. It suggests that our proposed method can use a single model to directly score images according to various attributes indicated in the caption. Without needing antonym prompts or pre-trained text embedding as anchors, our model learns quantitative knowledge from the instructions alone.

## 4.4 ABLATION STUDIES

Our ranking-aware attention CLIP stands out by focusing on text-driven visual differences to score images. In this section, we examine two key components of our ranking adapter: the auxiliary ranking branch and the design of the ranking-aware module.

**Effect of the auxiliary ranking branch.** We validate whether additional supervision from visual differences enhances ranking performance. We compare three settings: (1) using only the regression head, (2) adding a separate ranking head with embedding subtraction ($z_i' - z_j'$) for ranking supervision, and (3) applying ranking-aware attention to the paired embedding ($z_{ij}'$) for the supervision.

As shown in Table 5, the regression head alone performs reasonably well (e.g., $0.402$ MAE on HCI and $0.612$ PLCC on object count sorting), demonstrating that reframing the ranking tasks as a learning-to-rank problem is effective compared to standard contrastive learning used for fine-tuning CLIP. Adding a separate ranking head for pairwise supervision yields noticeable improvements, particularly on the HCI dataset, highlighting the value of additional ranking supervision. Notably, its improvement is limited in complex tasks like object count sorting, where the model requires to focus on the queried object within a cluttered image for image scoring. Incorporating ranking-aware attention in the ranking branch further enhances performance, with a $+21.14\%$ gain on HCI, $+1.96\%$ in PLCC, and $+3.53\%$ in SRCC for object counting. These results indicate that additional ranking supervision benefits image ranking tasks.

Table 5: **Ablation study on the design of the ranking adapter.** The first row stands for fine-tuned CLIP, serving as a reference.

| | | | HCI | Object count sorting | |
|---|---|---|---|---|---|
| LTR paradigm | Ranking Head | Ranking-aware Attention | MAE ($\downarrow$) | PLCC ($\uparrow$) | SRCC ($\uparrow$) |
| - | - | - | 1.113 | 0.251 | 0.422 |
| $\checkmark$ | - | - | 0.402 | 0.612 | 0.538 |
| $\checkmark$ | $\checkmark$ | - | 0.355 (+11.69%) | 0.619 (+1.14%) | 0.536 (−0.37%) |
| $\checkmark$ | $\checkmark$ | $\checkmark$ | 0.317 (+21.14%) | 0.624 (+1.96%) | 0.557 (+3.53%) |

Table 6: **Ablation study on the components of ranking-aware attention.**

| | | | HCI | Object count sorting | |
|---|---|---|---|---|---|
| Component | Used | Ablated | MAE ($\downarrow$) | PLCC ($\uparrow$) | SRCC ($\uparrow$) |
| Ours | - | - | **0.317** | **0.624** | **0.557** |
| (1) Attention output computation | $O_i = A_i \cdot V_i; O_j = A_j \cdot V_j$ | $O = A \cdot V$ | 0.355 | 0.602 | 0.542 |
| (2) Output representation | $O_{i,j} = O_i - O_j$ | $O_{i,j} = [O_i; O_j]$ | 0.347 | 0.609 | 0.540 |
| (3) Attention mechanism | Cross attention | Self attention | 0.339 | 0.621 | 0.545 |

**Ablation study on the design of ranking-aware attention.** Several designs can potentially enable the attention module to capture the visual differences between image pairs. We validate our design by swapping components within the ranking-aware module individually.

In Table 6, we ablate (1) the split dot-product ($A_i \cdot V_i - A_j \cdot V_j$) by merging it into $A \cdot V$; (2) We compare explicit subtraction with concatenate for embedding fusion: ($O_i - O_j$ vs. $[O_i; O_j]$); (3) We assess attentional pooling against self-attention for computing the attention matrix.

As shown in Table 6, both the split dot-product and explicit subtraction of output embeddings are essential, leading to improvements in MAE, PLCC, and SRCC. Using a merged dot-product for attention outputs may confuse the model about response sources, leading to performance degradation. Regarding the comparison between concatenation and explicit subtraction of output embeddings, our results show that explicit subtraction performs better, likely because it aligns more naturally with the ranking task. Finally, we observe that using cross-attention to aggregate responses from image pairs is more efficient regarding performance and GPU memory usage. In contrast, self-attention consumes about twice the GPU memory (due to the initial dot-product operation, $Q \cdot K$) and delivers worse performance. This result suggests that self-attention, which requires calculating relationships between all elements across image pairs, may lead to a higher computational burden and increased complexity, making training convergence more difficult.

## 5 CONCLUSIONS

This work presents an efficient and scalable framework for text-guided image ranking. By reframing CLIP's image-text contrastive learning into a learning-to-rank task and introducing a ranking-aware attention mechanism, our model effectively captures text-driven visual differences between image pairs without relying on task-specific fine-tuning or curated datasets. Experiments show the effectiveness of our approach, surpassing CLIP baselines and achieving results comparable to state-of-the-art models tailored for specific tasks. Overall, our work highlights the potential of integrating VLMs with ranking tasks, utilizing text-guided visual distinctions for the sense of quantitative concepts.

**Limitations and future works.** Our ranking module can be trained on diverse text prompts covering single and multi-task properties. For example, it can sort images based on criteria such as the number of cats and image quality with a single model. As shown in Table 10 (Appendix B.3), our approach achieves performance on par with models trained for individual tasks. While our method shows convincing results in ranking images by two attributes simultaneously (Appendix E), the lack of ground truth for multi-attribute rankings complicates evaluation. Besides, as we adapt CLIP to rank images by specified queries, our method retains its category-specific bias, hindering its performance on open-vocabulary tasks like counting "objects of whatever categories." The pairwise nature of our ranking-aware attention limits its compatibility with triplet-based ranking losses, such as the triplet or ranked list losses. Future research directions include exploring strategies and datasets for multi-attribute and open-vocabulary image ranking. Additionally, incorporating diverse ranking losses beyond pairwise comparisons could improve the model's robustness and adaptability.

## 6 ACKNOWLEDGMENT

This work was supported in part by the National Science and Technology Council (NSTC) under grants 112-2221-E-A49-090-MY3, 111-2628-E-A49-025-MY3, and 113-2634-F-006-002.

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

## A    EVALUATE ON OTHER DATASETS

### A.1    FACIAL AGE ESTIMATION ON THE UTKFACE DATASET.

Table 7 shows additional results on facial age estimation using UTKFace dataset (Zhang et al., 2017). Following the preprocessing and data split described in Kuprashevich & Tolstykh (2023), we train our ranking-aware adapter for $20k$ steps with a batch size of $64$, a learning rate of $5e - 5$, and a weight decay of $0.01$. Our method outperforms MiVOLO (Kuprashevich & Tolstykh, 2023), designed to predict a person's age based on facial or full-body images.

Table 7: **Facial Age Estimation Results on the UTKFace Dataset.** We take numbers from Kupra-shevich & Tolstykh (2023) for the results of the reference methods.

| Method | MAE ($\downarrow$) |
|---|---|
| CORAL | 5.39 |
| Randomized Bins | 4.55 |
| MWR | 4.37 |
| MiVOLO | 4.23 |
| Ours | **3.83** |

### A.2    OBJECT COUNT SORTING ON ON CLEVR DATASET.

The CLEVR dataset (Johnson et al., 2017) is often used to assess complex relational understanding, such as answering questions like "Are there an equal number of large things and metal spheres?" in a visual question-answering context. Here, we focus on a simple yet general counting task by counting instances that match the queried attribute. Specifically, we train and evaluate the model's ranking ability based on the counts of three attributes: color (8 colors), material (2 materials), and shape (3 shapes), using a single model. For example, we use the prompt "`Sort images by the number of objects in red.`" to order images by the number of red objects. To the best of our knowledge, no existing benchmark aligns with our setting, so we use CLIP and InstructBLIP (Dai et al., 2023) for comparison. As demonstrated in Table 8, our approach effectively ranks images based on the queried attributes, outperforming both the CLIP baseline and InstructBLIP. Figure 6 further illustrates the quantitative results for queries based on different attributes.

## B    MORE EXPERIMENTAL DETAILS

### B.1    OBJECT COUNT SORTING RELATIVE TO SPECIFIC COUNTS

Sorting images based on the counts of a queried object is a straightforward way to assess model performance. However, our method is viable for sorting images by relevance to other specified counts of a queried object.

To validate this, we conduct an experiment where, during training, we randomly sample a reference count $n$ and adjust the target values to reflect the distance between the actual object count and $n$. For instance, given a batch with images containing $[0, 3, 5, 10]$ dogs and a reference count of $4$, the target values for ranking become $[4, 1, 1, 6]$.

Figure 7 shows that with a query of "`0 dogs,`" images without dogs are ranked highest, followed by images with 2, 4, and 8 dogs. Similarly, for queries like "`3 dogs`" or "`6 dogs,`" the sorting results match our expectations.

### B.2    COMPARISONS WITH GENERAL VISION-LANGUAGE MODELS

General-purpose vision-language models (VLMs) have demonstrated strong capabilities in solving tasks via a visual-question-answering paradigm. In this study, we evaluate their performance on ranking tasks, specifically testing InstructBLIP (Dai et al., 2023) across all tasks and VILA-VLM (Ke et al., 2023) on the object count sorting task (Table 2). The text prompt used for each task is as follows:

Table 8: **Object Count Sorting Results on the CLEVR Dataset.**

| Method | Backbone | Color | | Material | | Shape | |
|---|---|---|---|---|---|---|---|
| | | PLCC | SRCC | PLCC | SRCC | PLCC | SRCC |
| CLIP (baseline) | ConvNext-L | 0.263 | 0.258 | 0.267 | 0.256 | 0.255 | 0.247 |
| InstructBLIP | ViT-g/14 | 0.194 | 0.170 | 0.332 | 0.315 | 0.584 | 0.548 |
| Ours | ConvNext-L | 0.992 | 0.836 | 0.992 | 0.981 | 0.993 | 0.966 |

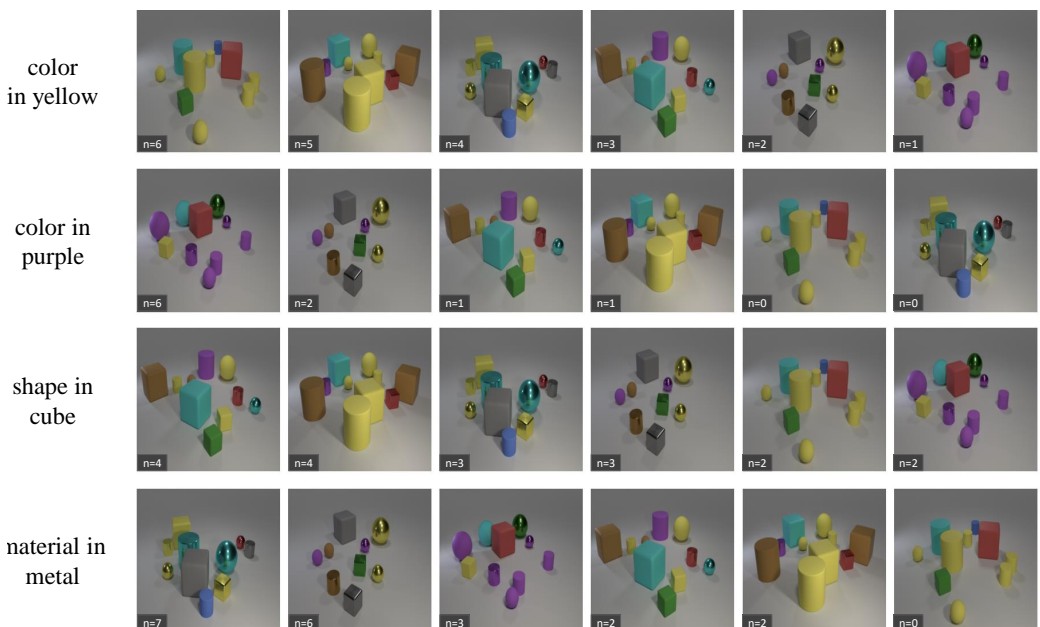

Figure 6: **Quantitative examples of querying objects by different attributes.** Images are ordered from high matching score to low matching score according to related query attributes.

- Object count sorting: "`How many {category} in the image?  Answer in a number.`"
- Image quality assessment:  "`How is the photography quality of the image?  Answer in a number from 1 to 10.`"
- Facial age estimation: "`How old is the person in the image?  Answer in a number.`"
- Historical colored image dating: "`When was this image taken (from 1930s to 1970s)?  Answer in a number.`"

For facial age estimation and HCI datasets, where ground truth labels are ordinal labels, we post-process the generated text into bins as defined by Wang et al. (2023d). As shown in Table 9, while InstructBLIP performs well in facial age estimation, it struggles with tasks such as image quality assessment and determining a photo's decade.

### B.3 TRAINING THE RANKING ADAPTER ACROSS MULTIPLE TASKS

Our method uses a learning-to-rank framework, leveraging text-driven visual differences for image ranking, which enables training across multiple tasks with diverse text queries. As shown in Table 10, the model trained on multiple tasks simultaneously achieves competitive performance compared to those trained on individual tasks. While there is a trade-off between task generalization and task-specific optimization — where diverse queries could slightly reduce individual task performance — this is balanced by the model's flexibility and its ability to handle multiple attributes across various

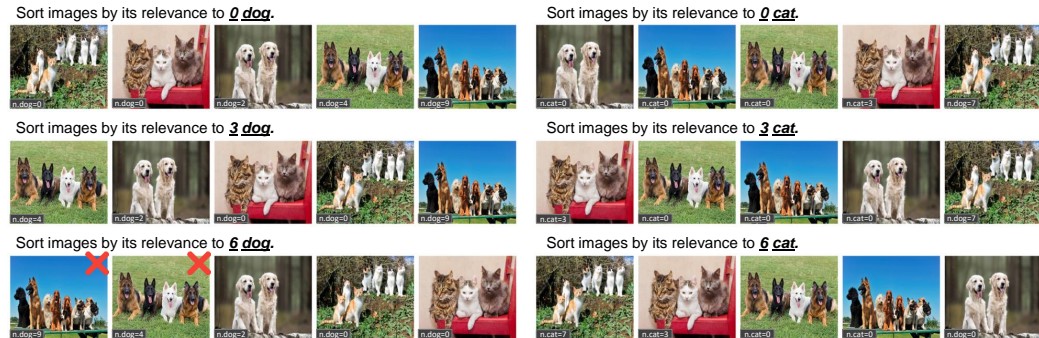

Figure 7: **Quantitative examples of queried by specific category relevant to specific counts.** Images are ordered from high matching score to low matching score with respect to the text query.

Table 9: **Comparison with general VLM methods.**

| | | Object count sorting | | KonIQA-10k | | Adience | HCI |
|---|---|---|---|---|---|---|---|
| Method | Backbone | PLCC | SRCC | PLCC | SRCC | MAE | MAE |
| CLIP (baseline) | ConvNext-L | 0.251 | 0.422 | 0.245 | 0.216 | 0.80 | 1.48 |
| InstructBLIP | ViT-g/14 | 0.509 | 0.485 | 0.211 | 0.163 | 0.41 | 0.96 |
| Ours | ConvNext-L | 0.624 | 0.557 | 0.919 | 0.911 | 0.36 | 0.32 |

Table 10: **Comparison between training on individual tasks and simultaneous training on multiple tasks.** Red and blue numbers denote the top two scores.

| | HCI | Adience | KonIQ-10k | | Object count sorting | |
|---|---|---|---|---|---|---|
| Training on | MAE ($\downarrow$) | MAE ($\downarrow$) | PLCC ($\uparrow$) | SRCC ($\uparrow$) | PLCC ($\uparrow$) | SRCC ($\uparrow$) |
| individual (HCI) | 0.31 | 1.50 | 0.013 | 0.030 | $-0.069$ | $-0.046$ |
| individual (Adience) | 2.25 | 0.36 | 0.044 | 0.061 | $-0.001$ | 0.036 |
| individual (KonIQ-10k) | 2.57 | 2.05 | 0.919 | 0.911 | 0.047 | 0.064 |
| individual (Object count sorting) | 2.99 | 4.09 | 0.044 | 0.065 | 0.624 | 0.557 |
| mixed | 0.36 | 0.43 | 0.881 | 0.876 | 0.657 | 0.575 |

Table 11: **Ablation for the different number of relational tokens in the ranking-aware module.**

| | HCI | KonIQ-10k | | Object count sorting | |
|---|---|---|---|---|---|
| #tokens | MAE | PLCC | SRCC | PLCC | SRCC |
| 1 | 0.321 | 0.9182 | 0.9104 | 0.6003 | 0.5211 |
| 4 | 0.343 | 0.9172 | 0.9087 | 0.6197 | 0.5483 |
| 9 | 0.321 | 0.9173 | 0.9087 | 0.6167 | 0.5451 |
| 16 | **0.313** | **0.9194** | **0.9107** | **0.6242** | **0.5571** |

domains. This offers a broader quantitative understanding across diverse scenarios and enables us to explore image ranking by multiple attributes simultaneously.

## B.4 EFFECT OF THE NUMBER OF RELATIONAL TOKENS IN RANKING-AWARE ATTENTION

Table 11 presents an ablation study on the impact of using different numbers of relational tokens $M$ in the ranking-aware attention module of our ranking adapter. The results indicate that $M = 16$ yields the best average performance, and we use this as the default setting for all other experiments.

Table 12: **Ablation for the different visual backbones and image resolution.**

| Backbone | #params | Image resolution | Object count sorting | | KonIQA-10k (MOS) | | HCI | Adience |
|---|---|---|---|---|---|---|---|---|
| | | | PLCC | SRCC | PLCC | SRCC | MAE | MAE |
| ViT-B/16 | 87M | 224 | 0.574 | 0.497 | 0.929 | 0.911 | 0.45 | 0.39 |
| ConvNext-B | 89M | 224 | 0.546 | 0.480 | 0.926 | 0.904 | 0.44 | 0.39 |
| ConvNext-L | 198M | 224 | 0.592 | 0.520 | 0.920 | 0.900 | 0.41 | 0.40 |
| ConvNext-L | 198M | 320 | 0.624 | 0.557 | 0.919 | 0.911 | 0.32 | 0.36 |

## B.5 EFFECT OF DIFFERENT VISUAL BACKBONE AND IMAGE RESOLUTION

Table 12 presents the effect of using different visual encoders and image resolutions among different tasks. The results demonstrate that our proposed method performs well across different backbones and image resolutions, with larger backbones and higher resolutions yielding improved performance.

## C CHALLENGES AND FAILURE CASES

Although the COCO-REM dataset refines the annotations of the COCO dataset, labeling densely packed objects remains challenging, often resulting in incorrect counts. A "crowded" flag is used to indicate densely packed objects. Figure 8 displays several challenging cases for our ranking adapter. Our model accurately assigns higher scores to images with a "crowded" flag for the object count sorting task, indicating accurate counts far exceeding the annotated counts. Such a "crowded" flag is typical in categories such as "Person", "Food", and "Animals", etc. This result highlights the advantage of using a ranking paradigm for quantitative comprehension, as it better tolerates noisy labels without directly pairing incorrect values with images. However, our model struggles with identifying icons or paintings as objects, such as cats painted on a bus or apples on a tablecloth, as in Figure 8(a).

In the image quality/aesthetics assessment task in Figure 8(b), the model's ranking results closely align with subjective "MOS" scores. Nonetheless, some images with similar content receive significantly different subjective scores, while our model will have similar scores to these images, such as the flower examples in Figure 8(b).

## D MORE QUALITATIVE EXAMPLES

As shown in Figure 9, we evaluate our ranking adapter on unseen categories using the LVIS dataset, which includes more categories than the COCO dataset. The model maintains its ability to rank target objects, such as butterflies, when the target categories are not coupled with others in the dataset. Performance drops drastically for categories like deer, which often appears with zebra or giraffe, or lemon, which is frequently found with apple or orange. Moreover, our model struggles with counting objects in images taken with a depth of field, such as those containing pillows.

For image aesthetics assessment, we apply our ranking adapter to the AGIQA-3k dataset, which includes generated artworks from various image generative models using the same prompt. As shown in Figure 10, we sample five images from the highest to lowest MOS at the same interval, finding that our model shows high agreement with subjective scores.

## E QUERY BY MULTIPLE ATTRIBUTES

Our proposed method can rank images on different tasks and with various attributes, making evaluating its performance with multiple attributes intriguing. However, combining two attributes into a single caption can cause issues: 1) it is difficult to determine which attribute the model focuses on more, and 2) how to determine the ground truth is unclear. Still, we explore the potential of our proposed method by evaluating images on the AVA dataset and AGIQA-3k dataset qualitatively. When combining multiple attributions, we use separate captions for each attribute to generate individual scores, normalize these scores, and then multiply them to rank images by both attributes. As shown in Figure 11, the top-5 and bottom-5 images retrieved by our method are highly reasonable. Additionally,

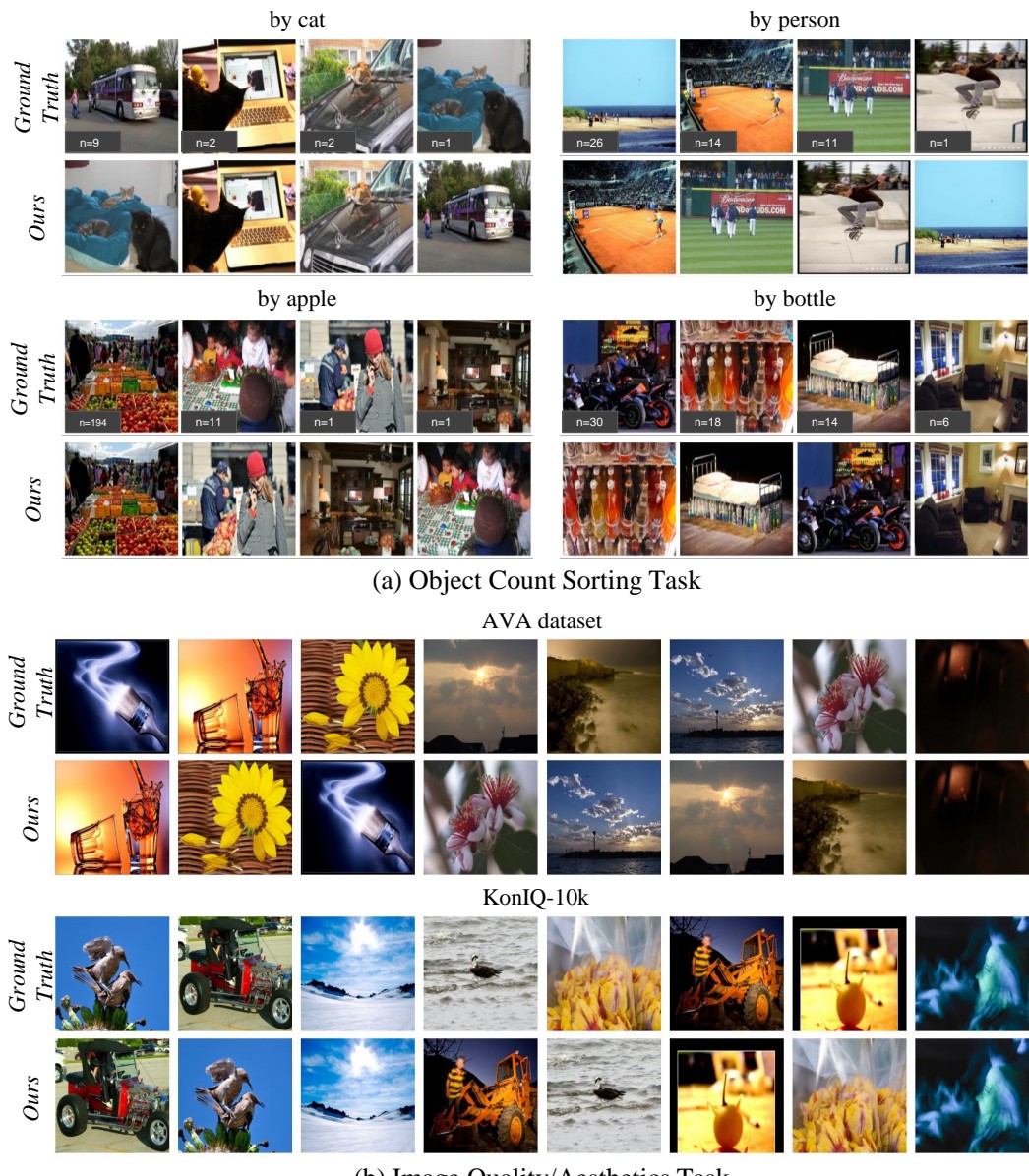

Figure 8: **Examples of challenging cases.**. The top row shows images ordered by annotated counts, and the bottom row shows images ordered by our model. a) For images in the COCO-REM dataset, icons (e.g., cat painting on the bus) may confuse the model, whereas it accurately scores crowded objects as seen in the examples of "*Person*", "*Apple*", and "*Bottle*". b) For the *MOS*, as the general order is similar, some images that have similar patterns (e.g., flowers) may have different preference scores.

when retrieving images based on an object category and an image property, the results align well with the query. By adjusting the weight given to different attributes, it is possible to tailor the retrieval results to favor one attribute over the other, highlighting the potential for future applications.

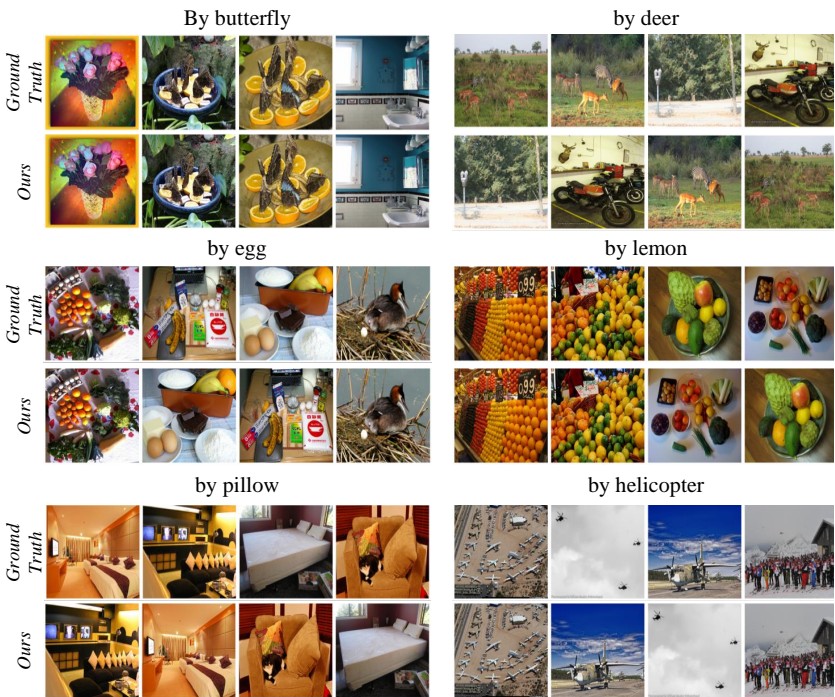

Figure 9: **Examples on the LVIS dataset (**Gupta et al., 2019**)**. Evaluation of our model on examples from unseen categories during training. The top row shows images ordered by annotated counts, and the bottom row shows images ordered by our model.

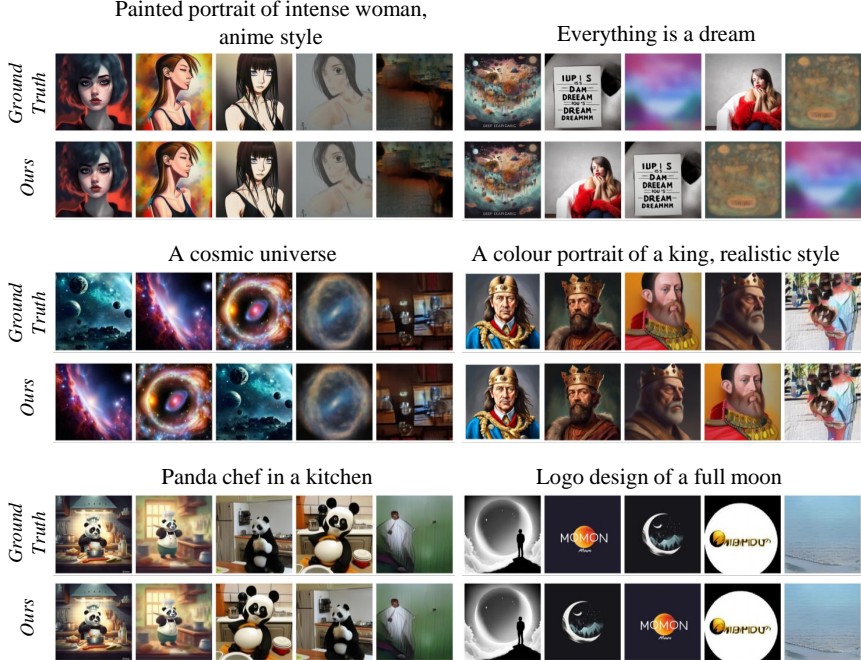

Figure 10: **Examples on AGIQA-3k (**Li et al., 2023a**)**. Comparison of AI-generated artworks in the AGI3Q dataset based on *MOS*. The top row shows images ordered by subjective scores from high to low, and the bottom row shows images ordered by our model.

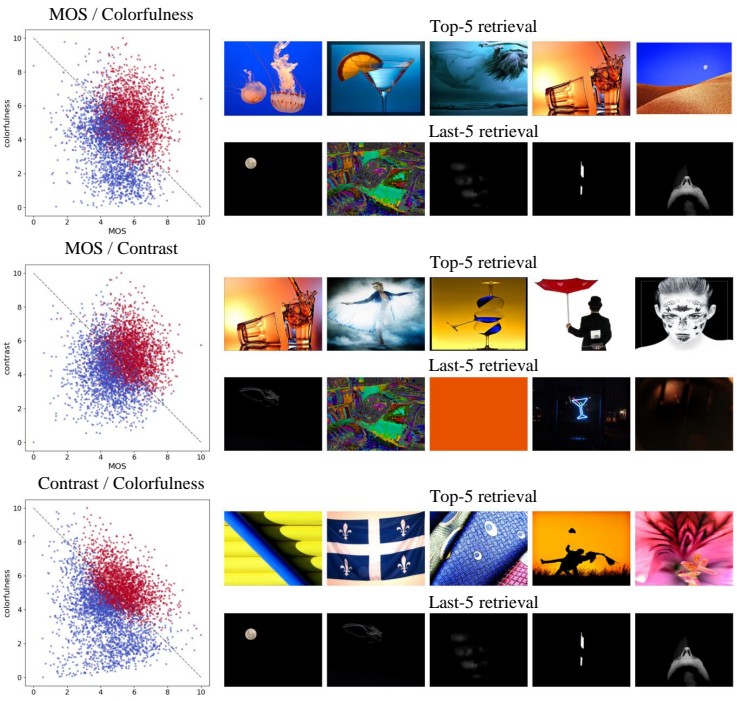

Figure 11: **Queried by multiple IQA attributes.** The scatter plot shows the normalized scores of two attributes, images with those normalized values above the median highlighted in red and those below in blue. Examples for the top-5 and last-5 retrievals of `MOS`/`Colorfulness`, `MOS`/`Contrast`, and `Contrast`/`Colorfulness`.

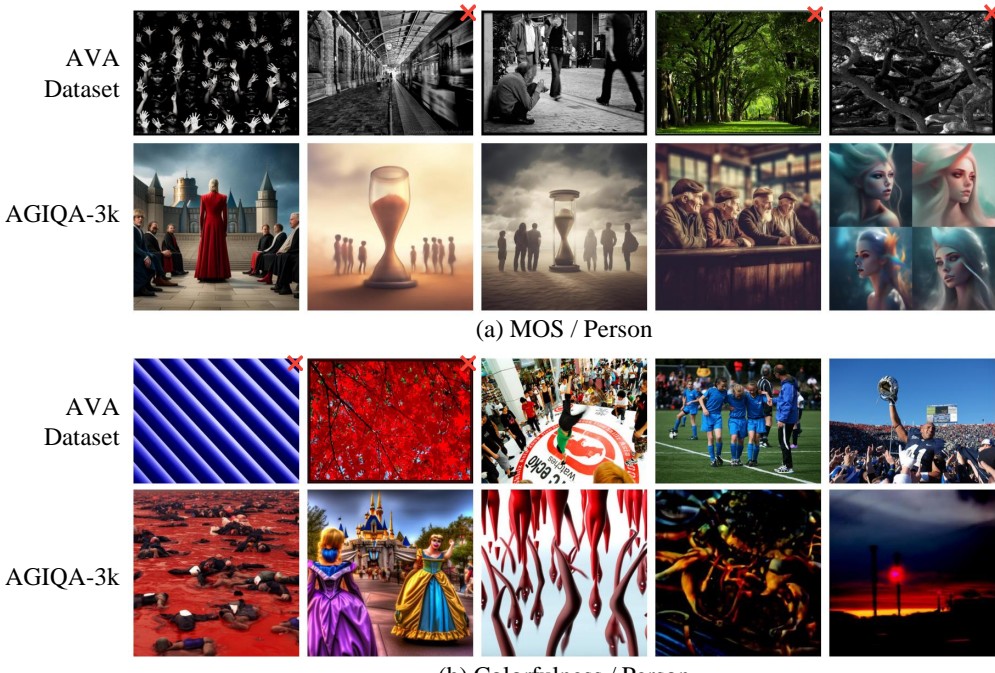

Figure 12: **Queried by multiple attributes across tasks.** Examples for the top-5 images retrieved using two attributes: (a) by `MOS` and by the number of "`person`" and (b) by `Colorfulness` and by the number of "`person`", on the AVA and AGIQA-3k datasets.

