# OpenReview forum: "Ranking-aware adapter for text-driven image ordering with CLIP"
_ICLR.cc/2025/Conference — ICLR 2025 Poster_

### Official Review · Reviewer_DJCM · 2024-10-23

**Soundness:** 4
**Presentation:** 4
**Contribution:** 4
**Rating:** 8
**Confidence:** 4

**Summary:**

This paper introduces a light-weight rank adapter which allows using an existing pre-trained CLIP model for any ranking task. The authors propose using learnable text prompts along with a light-weight cross-attention based ranking module which can measure visual differences between pairs of images. This approach is much simpler than some of the preceding works as it is able to be trained for the learning-to-rank task using a single text prompt such as "sort by number of [category]" compared to some of the prior works which would try to use multiple text prompts followed by contrastive learning to teach CLIP how to rank. Their solution is intuitive, elegant and can be readily adapted to any existing CLIP type framework without the need of fine-tuning the CLIP encoders. They show SoTA performance on a variety of learning-to-rank tasks such as facial age estimation, colored image dating, object counting etc. They also validate all their design choices in the ranking module with ablations and show abundant qualitative examples of their method. The appendix also covers interesting future directions such as ranking based on multiple attributes and analyzing how the transferability of ranking works across domains. Overall I think the paper is quite well written and provides a very elegant solution to the learning to rank problem and is well substantiated with experiments and ablations.

**Strengths:**

1) The paper is very well written and it is clear to understand the method section. The idea of replacing multiple rank specific prompts in prior work with a single 'sort by x' is simple and well motivated and simplifies the architecture quite a bit. The use of cross-attention in both the ranking adapter and the auxiliary ranking module makes intuitive sense to obtain text-conditioned image features and to get "pairwise image" differences respectively.
2) The experimental section is well detailed and results are shown on a variety of different benchmarks for ranking. The appendix also covers more examples of tasks. The ablation section also validates the design choices for the ranking modules quite well. Overall the results are impressive beating all existing methods on a wide range of tasks.
3) The method proposed is light-weight as it does not require fine-tuning of CLIP encoders and only of the ranking modules and all the experiments were run on a single 3090Ti GPU, which is impressive.
4) The paper discusses abundant qualitative examples across the different benchmarks and also discusses interesting future directions such as ranking across multiple attributes and the generalizability of ranking across benchmarks.

**Weaknesses:**

1) The paper only shows results on a single image backbone in the CLIP architecture (ConvNeXt-L). Also, it resizes images to 320X320 whereas most of the prior work (LR-CLIP, Ordinal-CLIP, NumCLIP) uses an image size of 224X224 and they use the ViTB/16 image encoder. It would be good to show results with this setting as it will ensure a fair comparison with prior baselines and it will also provide some insight into the generalizability of the ranking modules across different architecture types.
2) The paper only shows results with the pairwise ranking loss. It might be interesting to look at how the results vary by using other losses like triplet loss or the ranked list loss as presented by Wang et al (2021). Using a ranked list loss might also allow to not use the auxiliary ranking module which computes the pairwise attention matrix for the image features.

**Questions:**

1) Have you tried any experiments in ranking images by the number of objects in general, irrespective of the category of objects? For example just sorting on the "number" of whatever object is present in the image. That might help gauge how well the architecture is able to abstract out the "count" of objects.

---

> ### Author Response · Authors · 2024-11-23
>
> Thanks for your encouragement and insightful suggestions. We have included additional experiments and organized responses to address individual questions below.
>
> **[Q1] Different backbone and image resolution choices**
>
> Thanks for pointing this out and we conducted more experiments to explore the effect of different backbones for fair comparisons.
>
> For object count sorting, we have already compared our method to the CLIP-based baselines, using identical settings (Table 2 of the main paper). In addition, we included InstructBLIP as a benchmark to evaluate the performance of a large vision-language model trained on extensive datasets with an extra-large backbone. We also explored the impact of different visual backbones and resolutions on our method. As reported in the table below, our method consistently achieves favorable performance compared to the CLIP baseline and InstructBLIP.
>
> | **Method**         | **Vision backbone** | **#params** | **Image resolution** | **PLCC (Object count sorting)** | **SRCC (Object count sorting)** |
> |-----|-----|-----|:-----:|:-----:|:-----:|
> | CLIP (baseline)     | ConvNext-L          | 198M        | 320  | 0.251 | 0.422 |
> | InstructBLIP        | ViT-g/14            | 1011M       | 224 | 0.509  | 0.485  |
> | Ours  | ViT-B/16            | 87M         | 224 | 0.574  | 0.497  |
> | Ours  | ConvNext-B          | 89M         | 224  | 0.546  | 0.480  |
> | Ours  | ConvNext-L          | 198M        | 224  | 0.592  | 0.520  |
> | Ours  | ConvNext-L          | 198M        | 320   | 0.624  | 0.557  |
>
> For the image quality assessment, facial age estimation, and HCI tasks, we conducted experiments with different vision backbones and image resolutions to compare our method with other approaches in the following table.
>
> | **Method**         | **Backbone**    | **#backbone params** | **Image resolution** | **Task-specific pretraining** | **PLCC (KonIQA-10k)** | **SRCC (KonIQA-10k)** |
> |------|-----|:-----:|:-----:|:-----:|:-----:|:-----:|
> | CLIP (baseline)     | ConvNext-L      | 198M                 | 320                  |     | 0.245                  | 0.216   |
> | CLIP-IQA+       | ViT-B/32        | 88M                  | 224                  |     | 0.895                  | 0.909  |
> | VILA-R          | ViT-B/16        | 87M                  | 224                  | ✓                             | 0.919                  | 0.932   |
> | Ours  | ViT-B/16        | 87M                  | 224                  |     | 0.929  | 0.911 |
> | Ours  | ConvNext-B      | 89M                  | 224                  |     | 0.926   | 0.904   |
> | Ours  | ConvNext-L      | 198M                 | 224                  |     | 0.920  | 0.900   |
> | Ours  | ConvNext-L      | 198M                 | 320                  |     | 0.919  | 0.911  |
>
> | **Method**       | **Backbone**   | **# backbone params** | **Image resolution** | **Task-specific module** | **MAE (Adience)** | **MAE (HCI)** |
> |-----|-----|:-----:|:-----:|:-----:|:-----:|:-----:|
> | CLIP (baseline) | ConvNext-L    | 198M                  | 320                  |                          | 0.80              | 1.48          |
> | OrdinalCLIP | ViT-B/16      | 87M                   | 224                  |                          | 0.47              | 0.67          |
> | L2RCLIP | ViT-B/16      | 87M                   | 224                  | ✓                        | 0.36              | 0.43          |
> | Ours   | ViT-B/16      | 87M                   | 224                  |                          | 0.39              | 0.45          |
> | Ours   | ConvNext-B    | 89M                   | 224                  |                          | 0.39              | 0.44          |
> | Ours   | ConvNext-L    | 198M                  | 224                  |                          | 0.40              | 0.41          |
> | Ours   | ConvNext-L    | 198M                  | 320                  |                          | 0.36              | 0.32          |
>
> As shown in the table, using different backbones (with the same image resolution) has a slight impact on performance, e.g., ViT-B/16 even achieves slightly better results than ConvNext-L on KonIQA-10k and Adience. For the HCI dataset, we find that it is more sensitive to the backbone choice and image resolution, in which it can be attributed by its small training dataset size (1250 images) that would require more careful model tunings.
>
> Nevertheless, our method still consistently outperforms the CLIP baselines and achieves competitive results against other methods that require a task-specific pre-training (e.g., VILA-R) or module (e.g., L2RCLIP). Specifically, L2RCLIP designs an additional module to learn ordinal representations from captions containing numerical information, such as "a photo of [1, 2, 3, ...] years old face.". While it can be effective for specific tasks, it cannot generalize to arbitrary scores, as required in tasks like image quality assessment (as discussed in Ln 53-84, 142-145).

---

> > ### Author Response · Authors · 2024-11-23
> >
> > **[Q2] Consideration of alternative ranking losses**
> >
> > We appreciate the reviewer's suggestions regarding alternative ranking losses. Nevertheless, our proposed ranking-aware attention, illustrated in Figure 3, is developed to extract relative responses from two images through Eq(3). Due to the pairwise nature of the ranking-aware attention, incorporating losses like triplet loss or ranked list loss would necessitate significant architectural modifications.
> >
> > Both triplet loss and ranked list loss explore relationships among triplets of instances, including an anchor, a positive instance, and a negative instance. Our ranking-aware attention, however, focuses on pairwise comparisons between images, making it incompatible with these losses.
> >
> > We agree that exploring alternative ranking losses is crucial for ranking tasks, e.g., optimization convergence and model stability. In the future, we will aim for a more flexible ranking-aware attention design to accommodate different losses.
> >
> > **[Q3] Sorting on the "number" of whatever objects present in the image**
> >
> > We acknowledge that this category-agnostic ranking task may pose challenges for our current method. First, both the vision and text encoders of CLIP are derived through a category-specific contrastive learning paradigm. Consequently, our model, rooted in CLIP, is inherently category-specific and struggles to perform category-agnostic object sorting. Second, sorting objects across arbitrary categories aligns closely with open-vocabulary object discovery, while our model is specifically adapted to queried object categories. Nonetheless, adapting pretrained VLMs to open-vocabulary ranking models presents an exciting direction for advancing query-based image ordering as one future work.
> >
> > As noted by the reviewer, we acknowledge the significance of this limitation and the potential of this future direction. We incorporate these points into the conclusion section of our paper as in Ln 535-539.

---

> > > ### Author Response · Authors · 2024-11-24
> > > **Please let us know whether we address all the issues**
> > >
> > > Dear reviewer,
> > >
> > > Thank you for the comments on our paper.
> > >
> > > We have submitted the response to your comments and updated our paper (PDF file). Please let us know if you have additional questions so that we can address them during the discussion period.
> > >
> > > Thank you

---

### Official Review · Reviewer_AayG · 2024-10-31

**Soundness:** 3
**Presentation:** 2
**Contribution:** 3
**Rating:** 6
**Confidence:** 4

**Summary:**

This paper introduces a modified CLIP model for text-guided image ranking, using a lightweight adapter with ranking-aware attention to enhance understanding of visual differences across multiple images. By incorporating learnable prompts for ranking instructions, the approach reduces dependence on extensive text prompting. The method outperforms fine-tuned CLIP on various tasks and rivals specialized models in areas like facial age estimation and image quality assessment.

**Strengths:**

1. The motivation is clear: while previous methods require generating multiple captions for input images, this approach only needs a single rank-related text prompt.
2. The proposed ranking-aware adapter achieves superior performance over fine-tuned CLIP and is competitive with specialized models for tasks like facial age estimation and image quality assessment, offering a versatile solution for text-guided image ranking.

**Weaknesses:**

1. Although the motivation of this paper is clear, the technical contribution does not appear strong enough from my perspective. I will wait to see other reviewers’ comments on this aspect.
2. In Equation (2), the symbols $V_i$ and $V_j$ are not explained. Adding the shapes or dimensions of certain symbols in Equation (2) would enhance clarity.
3. Regarding the experiments: why does ranking-aware attention utilize three MLP blocks?

Minor Issues:
- Consider adding count numbers to the results in Figures 5 and 6 for clarity, as done in Figure 7(a).
- The citation format in Lines 417, 309, 160, and 161 appears inconsistent. I suggest the authors review the full paper and adjust citation formatting for consistency with Line 424 if needed.
- The symbols between Figure 3 and the main text are inconsistent. For example, the symbol for relation tokens is $\mathbf{q}$ in Figures 2 and 3 but $q$ in Equation (1).
- Some symbols in Figure 3 are unclear. $A_i$ is a matrix, but $P$ represents dimension. What does $A_i(1-P)$ signify?
- The column margins in Tables 1 and 2 are too wide, causing the tables to appear cramped together, and the right edge of the table extends beyond the page, which affects visual appeal.

**Questions:**

- Are the weights of the relational ranking-aware attention also frozen? If they are trainable, a trainable symbol should be indicated in Figure 2.
- How does the relational ranking-aware attention handle multiple images (i.e., more than two)?

---

> ### Author Response · Authors · 2024-11-22
>
> Thanks for your insightful comments and suggestions.
>
> **[Q1] Technical contributions**
>
> Our proposed method focuses on designing a general framework for image ranking tasks guided by queried text prompts. We highlight our key contributions below:
> 1. We transform the image-text contrastive objective into a learning-to-rank objective in CLIP-based models, thereby allowing the model to **directly learn the differences across image instances, rather than aligning the image one-by-one with its numerical value as in the prior work** (see Figure 1).
> 2. We adapt the pretrained-CLIP with a lightweight ranking-aware adapter, which is a general design for various ranking tasks.
> 3. We propose to use two branches in our ranking-aware adapter: a regression branch for score prediction and an auxiliary ranking branch for learning image differences, which involves a relational ranking-aware attention module to capture text-conditioned visual differences across image pairs. Our ablation study (Table 5 of the main paper) demonstrates the effectiveness of these design choices.
>
> Overall, our proposed method does not rely on task-specific modules for ranking tasks: 1) unlike existing CLIP-based methods (Paiss et al., 2023; Wang et al., 2023d; Ln 81-83, 142-145), our approach is not required to use prompts for specific numbers (e.g., "one cat", "two cats", etc.) and perform image-number matching, and (2) unlike the VILA method (Ke et al., 2023; Ln 451-454) designed for IQA, our method does not need additional task-specific pretraining or post-processing.
>
> **[Q2] Formatting issues**
>
> We have carefully addressed the formatting issues in the revised paper, including texts, equations, figures/tables, and citations.
> - For the $V_{i}$ and $V_{j}$ in Eq(2), we revised the definition in the updated paper in Ln 253. The text-conditioned visual embeddings $z^{\prime}\_{i}$and $z^{\prime}\_{j}$ are utilized as values for computing the attention outputs, denotes as $V_{i} = z^{\prime}\_{i} \in \mathbb{R}^{p \times d'}$ and $V_{j} = z^{\prime}\_{j} \in \mathbb{R}^{p \times d'}$.
> - For Figure 5 and Figure 6, we added the counts to the image.
> - We revised the citation format in Ln 160-161, 186-187, 309, 319, 321, 416.
> - We aligned the format of symbol $q$ in Eq(1) and Figure 2/Figure 3.
> - We revised the symbol $P$ to $p$ to remove the confusion in Figure 3 and related contents as in Ln 243, 244, 252, and 354.
> - We fixed the table margin for Table 1 and Table 2.
>
> **[Q3] Numbers of MLP block in the ranking branch**
>
> Despite a minor effect, we determine the number of MLP blocks empirically based on the performance. We adopt the same setting across all tasks and experiments.
>
> **[Q4]: Weights of the relational ranking-aware attention**
>
> The relational ranking-aware attention module is trainable. We have updated Figure 2 to make it more clear.
>
> **[Q5]: Handle multiple images**
>
> Our relational ranking-aware attention module does handle multiple images by capturing text-conditioned visual differences across image pairs. For example, if there are three images, it will calculate the visual embedding difference of 3 pairs ($I_1$ vs. $I_2$, $I_1$ vs. $I_3$, and $I_2$ vs. $I_3$). These embedding differences will then align to their ground truth relationships and can be optimized by using the pairwise ranking loss as in Eq(5), based on the output embedding $O_{ij}$ of our attention module (Figure 3).

---

> > ### Author Response · Authors · 2024-11-24
> > **Please let us know whether we address all the issues**
> >
> > Dear reviewer,
> >
> > Thank you for the comments on our paper.
> >
> > We have submitted the response to your comments and updated our paper (PDF file). Please let us know if you have additional questions so that we can address them during the discussion period. We hope that you can consider raising the score.
> >
> > Thank you

---

> > > ### Comment · Reviewer_AayG · 2024-11-25
> > >
> > > Thank you very much for the author’s response. Most of my concerns have been addressed, and I will maintain my original score of borderline accept. However, I would like to further clarify one of my concerns regarding the symbols (i.e., $A_i(1-p)$) in Figure 3. In my view, the updated symbols remain somewhat confusing, as they can be misinterpreted as two matrices being multiplied, even though $(1-p)$ is not a matrix. I suggest using a different color (e.g., gray) to denote the dimension $(1-p)$ and avoiding placing $A_i$ and $(1-p)$ on the same line to improve clarity.

---

> > > > ### Author Response · Authors · 2024-11-26
> > > >
> > > > Thank you for your thoughtful suggestions to enhance the readability of Figure 3. We have updated the figure by using distinct colors to differentiate indices ($1 \sim p$) and ($p+1 \sim 2p$) and by splitting the lines for $A_{i}$ and $A_{j}$ to improve clarity.
> > > >
> > > > Once again, we thank you for all your insightful comments and suggestions.

---

### Official Review · Reviewer_FmxD · 2024-11-03

**Soundness:** 3
**Presentation:** 3
**Contribution:** 3
**Rating:** 8
**Confidence:** 4

**Summary:**

The paper proposes an efficient approach to enhance the CLIP model for image ranking tasks by reframing it as a learning-to-rank problem. It introduces a lightweight adapter with learnable prompts and a ranking-aware attention branch to leverage text-conditioned visual differences. This method consistently outperforms fine-tuned CLIP models across various tasks, achieving competitive results in facial age estimation and image quality assessment. By focusing on image ranking with a single instruction, the approach offers a generalized way to learn from visual differences without relying heavily on extensive text prompts.

**Strengths:**

1. The writing is clear and well-structured, making the content easy to follow. The logical flow helps readers grasp the key concepts without difficulty.

2. The motivation behind the study is explained exceptionally well. It highlights that existing research often centers on reasoning from a single image and relies heavily on text prompts. This approach restricts the ability to achieve a comprehensive understanding when multiple images are involved. By addressing these limitations, the study aims to enhance multi-image reasoning capabilities.

3. The method diagram is presented with clarity, making it easy for readers to comprehend the proposed approach. This visual aid effectively supports the explanation of complex processes, ensuring that the methodology is accessible to a broad audience.

4. The experimental results show impressive performance across several datasets. This demonstrates the robustness and effectiveness of the proposed methods, indicating their potential for broader application in various contexts.

**Weaknesses:**

1. Comparison with Existing Methods: There is a need to clearly delineate the core differences between OrdinalCLIP, L2RCLIP, and NumCLIP compared to existing methods, which might not be fully addressed.

2. State-of-the-Art Comparisons: The article does not adequately compare the proposed models to state-of-the-art multi-modal large language models (LLMs), which could provide a more comprehensive evaluation of their performance.

3. Performance on Complex Benchmarks: The performance on complex counting benchmarks like TallyQA and CLEVR is not thoroughly evaluated, leaving questions about the models' capabilities in more challenging scenarios.

**Questions:**

1. The paper utilizes CLIP with ConvNeXt-L, which differs from the backbone networks in existing methods. Does this affect the fairness of experimental comparisons?

2. Does the model require separate adapter networks for different tasks, and how might this impact its generalization to unseen scenarios?

---

> ### Author Response · Authors · 2024-11-22
>
> **[Q1] Core difference compared to the prior work (OrdinalCLIP, L2RCLIP, and NumCLIP)**
>
> As described in "Sec. 2 Learning-to-rank (Ln 142-145)" and Figure 1, **unlike these existing CLIP-based methods that align the image one-by-one with its numerical value (e.g., "2 cats", "3 cats", etc.), our method directly learns visual differences across image instances with a general text prompt**, e.g., "Sort images by the number of cats.".
>
> This is achieved by our proposed learning-to-rank framework with the ranking-aware adapter, which is composed of two branches that are adopted for various ranking tasks: a regression branch for score prediction and an auxiliary ranking branch for learning image differences. Our ablation study (the last three rows in Table5) validates the effectiveness of our design choices.
>
> **[Q2] Comparisons with general VLM methods**
>
> For the object count sorting task, we have compared our method to the VLM models in Table 2 (also in Ln 425-426), e.g., VILA-VLM (Lin et al., 2023).
>
> We also add another benchmark using InstructBLIP (Dai et al., 2023) for the object count sorting, image quality assessment (IQA), facial age estimation, and historical colored image dating (HCI) in the following table.
>
> | **Method**         | **Backbone**    | **#params** | **PLCC (Object count sorting)** | **SRCC (Object count sorting)** | **PLCC (KonIQA-10k)** | **SRCC (KonIQA-10k)** | **MAE (Adience)** | **MAE (HCI)** |
> |---------------------|-----------------|-------------|----------------------------------|----------------------------------|-----------------------|-----------------------|------------------|---------------|
> | CLIP (baseline)     | ConvNext-L      | 198M        | 0.251                            | 0.422                            | 0.245                 | 0.216                 | 0.80             | 1.48          |
> | InstrctBLIP         | ViT-g/14        | 1,011M      | 0.509                            | 0.485                            | 0.211                 | 0.163                 | 0.41             | 0.96          |
> | **Ours**            | ConvNext-L      | 198M        | **0.624**                        | **0.557**                        | **0.919**             | **0.911**             | **0.36**         | **0.32**      |
>
> For each task, the prompt for the InstructBLIP is:
> - Object count sorting: `How many {category} in the image? Answer in a number.`
> - Image quality assessment: `How is the photography quality of the image? Answer in a number from 1 to 10.`
> - Facial age estimation: `How old is the person in the image? Answer in a number.`
> - Historical colored image dating: `When was this image taken (from 1930s to 1970s)? Answer in a number.`
>
> For facial age estimation and HCI datasets, where ground truth labels are ordinal labels, we post-process the generated text into bins as defined by Wang et al. (2023d). As shown in the table, while InstructBLIP performs well in facial age estimation, it struggles with tasks such as image quality assessment and determining a photo's decade. In contrast, our proposed method performs better than the CLIP baseline under the same setting and the general VLM pretrained on large datasets.

---

> > ### Author Response · Authors · 2024-11-22
> >
> > **[Q3] Performance on the complex benchmark: CLEVR**
> >
> > We add experiments using the CLEVR dataset to evaluate our proposed method. While CLEVR is often used to assess complex relational understanding, such as answering questions like "Are there an equal number of large things and metal spheres?" in a visual question-answering context, we focus on a simple yet general counting task. Specifically, we train and evaluate the model's ranking ability based on the counts of three attributes: color (8 colors), material (2 materials), and shape (3 shapes), using a single model. For example, we use the prompt: `Sort images by the number of objects in red.` to order images by the number of red objects. To the best of our knowledge, no existing benchmarks align with our setting, so we use CLIP and InstructBLIP as baselines for comparisons.
> >
> > | **Method**        | **Backbone**    | **PLCC (Color)** | **SRCC (Color)** | **PLCC (Material)** | **SRCC (Material)** | **PLCC (Shape)** | **SRCC (Shape)** |
> > |--------------------|-----------------|------------------|------------------|---------------------|---------------------|------------------|------------------|
> > | CLIP (baseline)    | ConvNext-L      | 0.263            | 0.258            | 0.267               | 0.256               | 0.255            | 0.247            |
> > | InstructBLIP       | ViT-g/14        | 0.194            | 0.170            | 0.332               | 0.315               | 0.584            | 0.548            |
> > | **Ours**           | **ConvNext-L**     | **0.992**        | **0.836**        | **0.992**           | **0.981**           | **0.993**        | **0.966**        |
> >
> > As shown in the table, our approach effectively ranks images based on the queried attributes, outperforming both the CLIP baseline and InstructBLIP. Moreover, unlike general VLMs, which often require post-processing to handle irrelevant generated texts for ranking scores, our proposed method offers a straightforward and efficient way to adapt CLIP for image ranking without the need for additional processing. We also provide qualitative results, where a random set of images is ranked using various attributes as prompts. Detailed quantitative results are included in the updated paper (Sec. A.2 and Figure 6 in the Appendix).

---

> > > ### Author Response · Authors · 2024-11-22
> > >
> > > **[Q4] Different backbone choices**
> > >
> > > Thanks for pointing this out and we conducted more experiments for fair comparisons. For the object count sorting task, we already compare our method to the fine-tuned CLIP benchmark using the same backbone and settings (Table 2 of the main paper). In addition, we include InstructBLIP as a benchmark to evaluate the performance of a large vision-language model trained on extensive datasets with an extra-large backbone (check the above [Q2] Comparisons with general VLM methods).
> > >
> > > For the image quality assessment, facial age estimation, and HCI tasks, we conduct experiments with different vision backbones and image resolutions to compare our method with other approaches in the following table.
> > >
> > > | **Method**         | **Backbone**    | **#backbone params** | **Image resolution** | **Task-specific pretraining** | **PLCC (KonIQA-10k)** | **SRCC (KonIQA-10k)** |
> > > |---------------------|-----------------|-----------------------|----------------------|-------------------------------|-----------------------|-----------------------|
> > > | CLIP (baseline)     | ConvNext-L      | 198M                 | 320                  |                               | 0.245                 | 0.216                 |
> > > | CLIP-IQA+       | ViT-B/32        | 88M                  | 224                  |                               | 0.895                 | 0.909                 |
> > > | VILA-R          | ViT-B/16        | 87M                  | 224                  | ✓                             | 0.919                 | 0.932                 |
> > > | Ours            | ViT/B-16        | 87M                  | 224                  |                               | 0.929                 | 0.911                 |
> > > | Ours            | ConvNext-B      | 89M                  | 224                  |                               | 0.926                 | 0.904                 |
> > > | Ours            | ConvNext-L      | 198M                 | 224                  |                               | 0.920                 | 0.900                 |
> > > | Ours            | ConvNext-L      | 198M                 | 320                  |                               | 0.919                 | 0.911                 |
> > >
> > > | **Method**         | **Backbone**    | **# backbone params** | **Image resolution** | **Task-specific module** | **MAE (Adience)** | **MAE (HCI)** |
> > > |---------------------|-----------------|:------------------------:|:----------------------:|:---------------------------:|:-------------------:|:---------------:|
> > > | CLIP (baseline)     | ConvNext-L      | 198M                  | 320                  |                           | 0.80              | 1.48          |
> > > | OrdinalCLIP     | ViT-B/16        | 87M                   | 224                  |                           | 0.47              | 0.67          |
> > > | L2RCLIP         | ViT-B/16        | 87M                   | 224                  | ✓                         | 0.36              | 0.43          |
> > > | Ours            | ViT-B/16        | 87M                   | 224                  |                           | 0.39              | 0.45          |
> > > | Ours            | ConvNext-B      | 89M                   | 224                  |                           | 0.39              | 0.44          |
> > > | Ours            | ConvNext-L      | 198M                  | 224                  |                           | 0.40              | 0.41          |
> > > | Ours            | ConvNext-L      | 198M                  | 320                  |                           | 0.36              | 0.32          |
> > >
> > > As shown in the table, using different backbones (with the same image resolution) has a slight impact on performance, e.g., ViT-B/16 even achieves slightly better results than ConvNext-L on KonIQA-10k and Adience. For the HCI dataset, we find that it is more sensitive to the backbone choice and image resolution, which can be attributed to its small training dataset size (1250 images) that would require more careful model tunings.
> > >
> > > Nevertheless, our method still consistently outperforms the CLIP baselines and achieves competitive results against other methods that require a task-specific pre-training (e.g., VILA-R) or module (e.g., L2RCLIP). Specifically, L2RCLIP designs an additional module to learn ordinal representations from captions containing numerical information, such as "a photo of [1, 2, 3, ...] years old face.". While it can be effective for specific tasks, it cannot generalize to arbitrary scores, as required in tasks like image quality assessment (as discussed in Ln 53-84, 142-145).
> > >
> > > **[Q5] Does our method require a separate adapter for different tasks?**
> > >
> > > Our ranking-awre adapter is task-agnostic, designed to rank images based on any given ranking-related text prompt. Adapting to a new scenario requires only the collection of relevant data with ranking relationships and re-training the model accordingly. The above CLEVR benchmark results shown in [Q3] present one example.

---

> > > > ### Author Response · Authors · 2024-11-24
> > > > **Please let us know whether we address all the issues**
> > > >
> > > > Dear reviewer,
> > > >
> > > > Thank you for the comments on our paper.
> > > >
> > > > We have submitted the response to your comments and updated our paper (PDF file). Please let us know if you have additional questions so that we can address them during the discussion period. We hope that you can consider raising the score.
> > > >
> > > > Thank you

---

> > > > > ### Comment · Reviewer_FmxD · 2024-11-25
> > > > >
> > > > > Thank you for your detailed response. My concerns have been fully addressed, and I have updated my evaluation to accept the manuscript.

---

### Official Review · Reviewer_a2Bs · 2024-11-03

**Soundness:** 2
**Presentation:** 1
**Contribution:** 2
**Rating:** 3
**Confidence:** 2

**Summary:**

The paper introduces a **Ranking-Aware Adapter for Text-Driven Image Ordering with CLIP**, designed to enhance CLIP’s capability in text-guided image ranking. Key points include:

1. **Objective**: The method reframes CLIP into a learning-to-rank (LTR) task, using a lightweight adapter to improve ranking tasks across image attributes (e.g., facial age, object count, image quality).

2. **Ranking Adapter Design**: The adapter consists of:
   - **Learnable Prompts** for adapting CLIP to new ranking instructions.
   - **Auxiliary Ranking Branch** that uses ranking-aware attention to focus on text-conditioned visual differences.
   - **Two Parallel Heads** for regression (individual image ranking score) and pairwise ranking supervision (to capture relative differences across images).

3. **Performance**: The adapter consistently outperforms fine-tuned CLIP on tasks such as facial age estimation, image dating, image quality assessment, and object count sorting.

4. **Evaluation and Ablation Studies**: The model is validated on multiple benchmarks, showing improved ranking accuracy, particularly as dataset complexity grows. Ablation studies highlight the contributions of each component, confirming the benefit of ranking-aware attention for pairwise comparisons.

This method enhances CLIP’s ability to rank images without exhaustive task-specific tuning, aiming to generalize across ranking tasks. Let me know if you'd like to dive into specific aspects like strengths, weaknesses, or contributions.

**Strengths:**

- *Generalizable Across Ranking Tasks*: The ranking-aware adapter is designed to handle multiple text-driven ranking tasks (e.g., age estimation, object counting, image quality assessment) without requiring extensive task-specific tuning. This flexibility suggests that the approach could generalize across diverse ranking applications, potentially making it adaptable for other vision-language tasks where relative comparisons matter.

- *Improved Performance on Benchmarks*: the proposed method improves performance on tasks like object count sorting, facial age estimation, and image aesthetics assessment, which could indicate its effectiveness in capturing ranking relationships across different domains.

**Weaknesses:**

- **Clarity and Reproducibility Issues**: The paper is challenging to follow, with several instances where the context is unclear, and symbols (e.g., Eq(5) and ΔO) are introduced without proper explanation. This lack of clarity makes it difficult to fully understand the method and poses challenges for reproducing the results. In particular, additional context is recommended regarding:
   - The specific role and application of the ranking score across different tasks.
   - Which of the two heads (regression or ranking) produces the final output, especially for each individual task.
   - How MAE is computed in experiments like facial age estimation and image dating, given that the model primarily performs ranking rather than explicit regression or classification.
   - What exactly is being ranked in each experiment, as this varies by task and isn’t sufficiently explained.

- **Ambiguity in Terminology and Methodology**: The claim that the paper “reframes the CLIP model into a learning-to-rank task” is ambiguous and could be misleading. A model cannot be reframed into a task; it can be adapted or extended to handle a task. In my understanding, the approach here is simply to add an adapter on top of the existing CLIP model, with CLIP’s image and text encoders used as frozen feature extractors. The authors are encouraged to clarify what they aim to achieve in reframing CLIP for ranking and to describe more precisely how this method differs from other approaches to ranking with CLIP features.

**Questions:**

Please see the weaknesses section on questions.

---

> ### Author Response · Authors · 2024-11-22
>
> Thanks for your valuable feedback and suggestions on the clarity of the paper. We respond to individual questions below.
>
> **[Q1] Symbol in Eq(5)**
>
> In Eq(5), $\Delta{O} = O_{i} - O_{j}$ which is $O_{i,j}$ defined in Eq(3). While the symbol $\Delta{O}$ is identical to $O_{ij}$, we remove $\Delta{O}$ to keep it consistent, and the paper is revised accordingly in Ln 273-275 of the updated paper.
>
> **[Q2] a) Role and application of the ranking score; b) Which of the two heads produces the final output? c) What exactly is being ranked in each task?**
>
> As mentioned in "Sec. 3.2 - Training objective and loss functions" (starting from Ln 264), the final output, i.e., the ranking score, is produced by the regression head for each image and thus the regressed scores can be used for ordering images in each task. On the other hand, the role of our proposed auxiliary ranking branch with relational ranking-aware attention is to enhance the representation by focusing on text-guided pairwise visual differences across images. As shown in our ablation study (the last three rows in Table 5), adding this auxiliary ranking branch improves the performance.
>
> For the target to be ranked in each task, we have described them in "Sec. 4.1" and also listed below:
> - Facial age estimation: the task is to rank images by the person's age group (8 groups). The prompt example is `Sort images by
> the person’s age group.`.
> - Historical colored image dating (HCI): the task is to rank images by the photo's taken date. The date category spans from 1930s to 1970s, corresponding to 1 to 5 in the experiment. The prompt example is `Sort images by the taken date of the photo.`.
> - Image quality/aesthetics assessment: the task is to rank the image by the photography quality, subjective preference, and objective properties like contrast and colorfulness. The values are normalized to 0.0 to 10.0. The prompt example is `Sort images by the image quality of colorfulness.`.
> - Object count sorting: the task is to rank images by the count of the queried object category. The value spans from 0 to more than a hundred. One prompt example is `Sort images by the number of cat.`.
>
> **[Q3] Computation of mean absolute error (MAE)**
>
> While the regression head produces the ranking score, we follow the setting as in previous studies (Wang et al., 2023d) to compute the distance between ground truths and predicted scores for fair comparisons with the prior work. Specifically, the facial age in the Adience dataset is partitioned into 8 ordinal age groups with an index spanning from 1 to 8; the historical color image dating dataset has images spanning from 1930s to 1970s with an index spanning from 1 to 5. For computing the MAE, we simply compute the distance between the predicted ranking score and the index of the image related to the text query.
>
> **[Q4] Ambiguity in Terminology and Methodology**
>
> We use the term "reframe" to describe the transformation of CLIP’s paired image-text contrastive objective into a ranking task with a learning-to-rank objective. This transformation allows the model to **directly learn visual differences across image instances, rather than matching a numerical value to an image as in existing CLIP-based methods** (see Figure 1). As the "reframe" can potentially lead to confusion, we will rephrase the term to improve the clarity as the reviewer suggests.
>
> Our paper adapts a pretrained CLIP model to rank images based on textual input, such as sorting images by properties like image conditions or quality. The proposed approach consists of three key components that differ from existing methods:
> 1. Transform the image-text contrastive objective into a learning-to-rank objective.
> 2. Introduce a lightweight ranking-aware adapter that fuses text embeddings into image embeddings and involves two branches: a regression branch for score prediction and an auxiliary ranking branch for learning image differences.
> 3. Develop a relational ranking-aware attention mechanism to capture text-conditioned visual differences across image pairs guided by their ranks.
>
> Unlike existing methods that (1) align the image with its number, such as CLIP-based approaches that use image-number matching (Paiss et al., 2023; Wang et al., 2023d; Ln 81-83, 142-145), or (2) require specialized architectures with task-specific pretraining, like VILA (Ke et al., 2023; Ln 451-454), our method is a general framework and does not require additional task-specific pretraining or post-processing, and is optimized directly through learning visual differences across image pairs. We validate our method on diverse ranking tasks, showing favorable performance compared to CLIP-baselines and competitive results against state-of-the-art methods designed for each individual task.

---

> > ### Author Response · Authors · 2024-11-24
> > **Please let us know whether we address all the issues**
> >
> > Dear reviewer,
> >
> > Thank you for the comments on our paper.
> >
> > We have submitted the response to your comments and updated our paper (PDF file). Please let us know if you have additional questions so that we can address them during the discussion period. We hope that you can consider raising the score.
> >
> > Thank you

---

> > > ### Comment · Reviewer_a2Bs · 2024-11-25
> > > **More Questions and Difficulties for Reproduction**
> > >
> > > First, I thank the authors for their detailed responses. However, I believe the clarity and reproducibility issues remain unresolved. Below are the key points of confusion:
> > >
> > > - In Eq. (1), the SoftMax operation normalizes over the concatenation of tokens from two images. In Eq. (2), the resulting attention scores are split into submatrices, and these are used to compute products with $ V_i $ and $ V_j$. However, it seems the submatrices are not properly normalized, raising doubts about whether the products $ O_i = A_iV_i $ and $ O_j = A_jV_j $ can be considered valid attention-weighted combinations. Consequently, the subtraction $ O_{i,j} = O_i - O_j $ lacks clear physical meaning and might fail to represent differential properties effectively. For instance, when $A_j = \mathbf{0} $ (all weights allocated to $ z_j $ in SoftMax), $O_{i,j} $ contains no information about $ z_i $, which undermines the method's reliability.
> > >
> > >
> > > - In the revised text, it is mentioned that the pairwise ranking branch is learned to predict $ O_i > O_j$. However, according to Eq (1)-(3), $ O_i $ and $O_j $ are vectors of dimension $ d $, and there are $ M $ such pairs. The text does not specify how vectors are compared (e.g., by norm or element-wise) or which pair among the $ M $ pairs is considered for the comparison. This lack of clarity adds significant confusion.
> > >
> > >
> > > - The definition of ranking scores remains unclear. The paper focuses on ordering images.  For using ranking scores to order images, any scale that preserves relative ordering (e.g., 0.1, 0.2, 0.3 vs. 1000, 2000, 3000) would suffice. Thus, it is critical to detail how the ranking score ground truth is derived and kept consistent across tasks. The absence of this explanation leaves the generalization between tasks ambiguous. Furthermore, the use of MAE as a metric for ranking scores seems valid only for ordinal classification tasks with a natural ground truth score. Without clarity on how this applies to the tasks described, confusion persists regarding the role of ranking scores and MAE.
> > >
> > > I hope these concerns can guide further clarification and improvements in the manuscript.

---

> > > > ### Author Response · Authors · 2024-11-26
> > > >
> > > > **[Q1] In Eq. (1), the SoftMax operation normalizes over the concatenation of tokens from two images. In Eq. (2), the resulting attention scores are split into submatrices, and these are used to compute products with $V_i$ and $V_j$. However, it seems the submatrices are not properly normalized, raising doubts about whether the products and can be considered valid attention-weighted combinations. $O_{i,j}=O_i - O_j$ lacks clear physical meaning. For instance $A_j=0$ ...**
> > > >
> > > > While $A_i$ and $A_j$ are not individually normalized, the produced $O_i$ and $O_j$ are designed to encode valuable information for ranking images $i$ and $j$.
> > > >
> > > > The justification is specified by considering the following two cases:
> > > > 1) $y_i > y_j$: As the query $q$ in Eq. (1) is expected to be more similar to tokens from image $i$, the elements of $A_i$ tend to be larger than those of $A_j$. This leads to $O_{i,j}$ in Eq. (3) encoding more features relevant to the query $q$ from image $i$. The reviewer's $A_j=0$ example is an extreme case, in which the features relevant to $q$ are completely from image $i$.
> > > > 2) $y_i < y_j$: In this opposite case, $O_{i,j}$ tends to encode the features relevant to the query $q$ from image $j$, but with a negative sign.
> > > >
> > > > By jointly considering the two cases, $O_{i,j}$ encodes features relevant to $q$ in a positive sign if $y_i > y_j$, and in a negative sign otherwise. Thus, $O_{i,j}$ provides informative evidence to rank images $i$ and $j$ conditioned on $q$. We will clarify the statement in the revised paper.
> > > >
> > > > To further evaluate the impact of individually normalizing $A_{i}$ and $A_{j}$, we conducted additional experiments on the Adience and HCI datasets, with results shown in the table below. Specifically, we applied the min-max normalization to the attention submatrices. Results indicate that this normalization step may not be critical for capturing the relative differences between image pairs.
> > > >
> > > > | **Method** | **Adience (MAE)** | **HCI (MAE)** |
> > > > | - | - | -
> > > > | **Current** | **0.36** | **0.32** |
> > > > | Individually nomalized $A_{i}$, $A_{j}$ | 0.39 | 0.35 |
> > > >
> > > > **[Q2] The pairwise ranking branch is learned to predict $O_i - O_j$. However, according to Eq (1)-(3), $O_i$ and $O_j$ are vectors of dimension $d$, .... The text does not specify how vectors are compared.**
> > > >
> > > > Thank you for pointing this out. Please note that in the two lines after Eq. (3) (i.e., Ln 259-260 of the revised paper), we state that $O_{i,j} = O_i - O_j$ is averaged over $M$ relational tokens and passed through a feed-forward network (FFN) to obtain the one-dimensional score for ranking images $i$ and $j$. Thus, after the processing in Ln 259-260, $O_{i,j}$ becomes a real value.
> > > >
> > > > We realize that it may cause unnecessary confusion. Therefore, we have revised Eq. (3) to $O_{i,j} = FFN(\frac{\sum_{m=1}^{M} (O_{i,m} - O_{j,m})}{M} )$, where $O_{i,m}$ is aggregated responses from image $i$ with the $m$-th relational token, and updated Figure 3 for clarity.
> > > >
> > > > **[Q3] a) Definition and computation of ranking score b) Generalization and output scales across tasks c) The use of MAE on evaluating certain tasks**
> > > >
> > > > The target for the regression branch is the ground truth value $y_{i}$ corresponding to the text query of the image defined for each task, e.g., "`3` cats" in the COCO-REM dataset, "The average subjective preference is `5.13.`" in the IQA task, or "a photo of person in age group `2`" as the facial age group in Adience. We follow each task's original ground truths as the ranking scores. While the scale may vary across tasks, fusing task-specific text embeddings into image embeddings ensures the output scale is appropriately handled by the guidance of text queries.
> > > >
> > > > As shown in our ablation study (Table 9, Appendix B.2), the model jointly trained on various tasks achieves competitive performance against models trained on individual tasks. This shows that our model is able to deal with various output scales but does not require ground truths with a consistent scale across tasks. As such, we can follow the prior work (OrdinalCLIP, L2RCLIP) to evaluate MAE on Adience and HCI for fair comparisons.

---

> > > > > ### Comment · Reviewer_a2Bs · 2024-11-27
> > > > >
> > > > > Thank you for clarifying $  O_{i,j} $ and the generation of ranking scores. I appreciate the effort to address these points, but I have the following additional comments and concerns:
> > > > >
> > > > > 1. While the stated goal of $ O_{i,j} $is to encode differential information between $ z_i $ and $  z_j $, the examples provided in the response highlight cases where only one side of the pair contributes to $ O_{i,j} $, effectively lacking the intended differential representation. This suggests that the underlying mechanism may not function as expected in certain scenarios. Additionally, the experimental results showing that normalizing  $ A_i $  and $ A_j $ makes no significant difference raise further questions. For example, in Table 6 row (2), the substitution of subtraction with concatenation ($ O_{i,j} = [O_i; O_j] $) also shows marginal impact on metrics. It also unknown whether using $O_i + O_j$ would change results. This observation challenges the importance of the "differential information" mechanism and suggests that it may not play a critical role in the method's performance. Such concerns, arising as the presentation becomes clearer, indicate that the paper may need further refinement before publication.
> > > > >
> > > > > 2. The updated Eq. (3) appears to significantly alter the method's implementation, potentially impacting the experimental results. Without clarity on whether the reported results correspond to the original or updated formulation, it is difficult to evaluate the validity of the findings. To resolve this uncertainty, I strongly recommend the authors provide the code implementation of the proposed method. This would not only verify the experimental setup but also enhance the reproducibility and transparency of the work.

---

> > > > > > ### Author Response · Authors · 2024-11-28
> > > > > >
> > > > > > **[Q1] The examples provided in the response highlight cases where only one side of the pair contributes to $O_{i,j}$, effectively lacking the intended differential representation. ...**
> > > > > >
> > > > > > We appreciate the reviewer for raising this interesting point about the encoded representation of $O_{i,j}$. Please note that the responses are contributed by both sides, not solely one coming from one side. As clarified in our earlier example, where $y_{i} > y_{j}$, we stated that:
> > > > > >
> > > > > > > as the query $q$ in Eq. (1) is expected to be more similar to tokens from image $i$, the elements of $A_i$ tend to be larger than those of $A_j$. This leads to $O_{i,j}$ in Eq. (3) **encoding more features** relevant to the query $q$ from image $i$.
> > > > > >
> > > > > > Here, We emphasize **"more"** features from image $i$ rather than "all" features, and thus this enables the encoding of relative differences between images. Moreover, during training, all images in a batch share the same rank-related text prompt but vary in task-specific values (Figure1 & 2). As a result, the attention submatrices $A_{i}$ and $A_{j}$ for any pair both encode query-related information, ensuring that $O_{i}$ and $O_{j}$ encode valuable information for ranking images. For the second case, where $y_{i} <y_{j}$, the interpretation is similar but with an opposite sign.
> > > > > >
> > > > > > Regarding the performance differences observed in the ablation study (Table 6) and the additional experiment on individually normalized attention submatrices mentioned earlier, swapping these ablated components in our relational ranking-aware module consistently yields worse results across tasks, indicating that the current design is a better choice.
> > > > > >
> > > > > > | **Method** | **HCI (MAE)** |
> > > > > > |---|:---:|
> > > > > > |$O_{i} - O_{j}$ | 0.32 |
> > > > > > |$O_{i} + O_{j}$ | 0.37 |
> > > > > >
> > > > > > **[Q2] The updated Eq. (3) appears to significantly alter the method's implementation?**
> > > > > >
> > > > > > We would like to clarify that **the implementation remains unchanged from the initial version to the current revision**; we have only enhanced the clarity of Eq. (3) by adding details already described in the original paper (Ln 259-260).
> > > > > >
> > > > > > In the previous Eq. (3) and its description in Ln 259-260,
> > > > > > > $O_{i,j} = O_i - O_j$, is averaged over $M$ relational tokens and passed through a feed-forward network (FFN) ...
> > > > > >
> > > > > > equals to
> > > > > >
> > > > > > > $O_{i,j} = FFN(\frac{\sum_{m=1}^{M} (O_{i,m} - O_{j,m})}{M} )$, where $O_{i,m}$ is aggregated responses from image $i$ with the $m$-th relational token, ...
> > > > > >
> > > > > > in our revised paper, which incorporates the description directly into the Eq. (3) for the improved clarity.
> > > > > >
> > > > > > To address your concerns, we will release the code and models upon acceptance.

---

> ### Author Response · Authors · 2024-12-02
>
> Dear Reviewer,
>
> Thank you for the comments on our paper. We have submitted the response to your comments. Please let us know if all issues have been addressed.
>
> Best, \
> Authors

---

### Meta-Review · Area_Chair_kAyF · 2024-12-11

**Metareview:**

The paper proposes to adapt CLIP for learning to rank (e.g. counting, rating) tasks, by optimizing differences in outputs. Extensive experiments demonstrate the effectiveness of the method. Common concerns include clarity of the presentation (method, terminology, reproducibility) and some experimental choices (methods compared, tasks included). These are addressed well and after the rebuttal phase, there is sufficient support to accept the paper.

**Additional Comments On Reviewer Discussion:**

Two reviewers did not provide final scores after engaging in discussion with the authors. One is these is the lowest-scoring reviewer, however this reviewer also states the lowest confidence among the four reviews. The other two reviewers comment that their concerns have been successfully addressed.

---

### Decision · Program_Chairs · 2025-01-22

Accept (Poster)